# Tropospheric Links to Uncertainty in Stratospheric Subseasonal Predictions

Rachel W.-Y. Wu[1], Gabriel Chiodo[1,4], Inna Polichtchouk[2], and Daniela I.V. Domeisen[3,1]

[1]Institute for Atmospheric and Climate Science, ETH Zurich, Zurich, Switzerland
[2]European Centre for Medium-Range Weather Forecasts, Reading, UK
[3]Faculty of Geosciences and Environment, University of Lausanne, Lausanne, Switzerland
[4]Instituto de Geociencias, IGEO-CSIC-UCM, Madrid, Spain

**Correspondence:** Rachel W.-Y. Wu (rachel.wu@env.ethz.ch)

**Abstract.** Variability in the stratosphere, especially extreme events such as Sudden Stratospheric Warmings (SSWs), can impact surface weather. Understanding stratospheric prediction uncertainty is therefore crucial for skillful surface weather forecasts on weekly to monthly timescales. Using ECMWF subseasonal hindcasts, this study finds that stratospheric uncertainty is most strongly linked to tropospheric uncertainty over the North Pacific and Northern Europe, regions that can modulate but also respond to stratospheric variability, suggesting a two-way propagation of uncertainty. A case study of the 2018 SSW event shows an initial poleward and upward propagation of uncertainty from tropical convection, followed by a downward propagation where ensemble members that accurately predict the SSW are also better at predicting its downward impacts. These findings highlight the locations in the troposphere that are linked to stratospheric uncertainty and suggest that improved model representation of tropospheric mechanisms linked to polar vortex variability could enhance both stratospheric and extratropical surface prediction.

## 1 Introduction

Anomalous variability in the stratosphere is an important precursor for surface weather anomalies (Baldwin and Dunkerton, 2001) and extremes (Domeisen and Butler, 2020) on weekly to monthly timescales in winter and spring. In particular, sudden stratospheric warming (SSW) (Baldwin et al., 2021) and strong vortex events are windows of opportunity for extended-range weather prediction (e.g. Domeisen et al., 2020b; Butler et al., 2018; Scaife et al., 2016). Indeed, the stratosphere has an extended predictability limit with respect to the troposphere (Domeisen et al., 2020a; Son et al., 2020). These longer characteristic timescales in the stratosphere suggest a potential for increased predictability of surface weather arising from stratospheric forcing, particularly on subseasonal-to-seasonal (S2S) timescales, ranging from weeks to months. However, when it comes to predicting the variability in the stratosphere in the first place, extreme stratospheric events, especially SSW events, have a more limited predictability as compared to more neutral states of the vortex. The average predictability of an SSW is around 5-10 days in dynamical models (Domeisen et al., 2020a; Taguchi, 2018; Chwat et al., 2022), indicating a higher uncertainty ahead of such events.

Uncertainty in the prediction of stratospheric variability can be contributed by model uncertainty in the stratospheric mean state and in upward wave propagation (Tripathi et al., 2015a), as the strength of the stratospheric polar vortex is modulated by the interaction of planetary waves with the stratospheric mean flow. The planetary waves entering the stratosphere can break, depositing momentum and thereby forcing a weakening of the westerly vortex winds. As a secondary effect, the breaking of planetary waves can also precondition the vortex into a state that is more favourable for wave propagation (Limpasuvan et al., 2004; Albers and Birner, 2014), which acts to guide waves towards the vortex (Matsuno, 1970), thus making the deposition of wave momentum more focused in the vortex area. Stratospheric variability can also be influenced by internal variability, where the stratosphere can be modulated by internal oscillations (e.g. Holton and Mass, 1976; Matthewman and Esler, 2011), or through amplifying wave activity that propagates from the troposphere (e.g. Clark, 1974; Plumb, 1981; Esler and Scott, 2005; Esler et al., 2006; Domeisen et al., 2018), which can lead to the triggering of SSWs even if the wave activity in the troposphere is not anomalous (Birner and Albers, 2017; de la Cámara et al., 2019). Hence, the strength and geometry of vortex winds and upward wave propagation can strongly influence the subsequent evolution of the polar vortex, and model biases related to these factors can therefore strongly impact the uncertainty in the prediction of the stratosphere.

Subseasonal-to-seasonal forecast systems are subject to model biases in both polar vortex strength (Lawrence et al., 2022) and in climatological tropospheric stationary waves (Schwartz et al., 2022), which can interact with wave anomalies to enhance or suppress upward wave flux (Smith and Kushner, 2012). For instance, the accurate representation of the vortex background state is found to be important for the successful prediction of the 2021 SSW event (Cho et al., 2023). Yet, it has been suggested that the dominant factor in limiting the prediction of SSWs is the prediction of planetary wave activity rather than the mean state (Stan and Straus, 2009; Wu et al., 2022; Portal et al., 2022). The major sources of uncertainty in predicting the wave activity driving SSWs are suggested to be associated with the model representation of tropospheric stationary wave ridges in western North America and the North Atlantic region (Schwartz et al., 2022). For individual SSW events, the uncertainty in wave activity is suggested to be related to the representation of extratropical blocking, as found for the 2018 SSW event (Karpechko et al., 2018; Lee et al., 2019; Statnaia et al., 2020) and to localized synoptic-scale tropospheric perturbations, as shown by Kent et al. (2023) for the 2013 SSW event.

Through teleconnection pathways, variability in the tropics can contribute to uncertainty in the extratropics, which can further propagate into the stratosphere (Straus et al., 2023; Roberts et al., 2023). The Madden-Julian Oscillation (MJO), the dominant mode of intraseasonal variability in the tropics, influences the extratropics by modulating extratropical tropospheric stationary waves, over the North Pacific in particular (Garfinkel et al., 2014; Lin et al., 2017; Schwartz and Garfinkel, 2017), and can further impact the stratospheric polar vortex by exciting poleward and vertical wave propagation (Garfinkel et al., 2012, 2014). Model initializations that better capture the MJO show better prediction skill over the North Pacific and Euro-Atlantic region (e.g. Ferranti et al., 2018; Kim et al., 2023) and better upward coupling of the troposphere to the stratosphere (Garfinkel and Schwartz, 2017; Stan et al., 2022), often resulting in a better simulation of SSWs (Schwartz and Garfinkel, 2020; Kang and Tziperman, 2018).

Uncertainty in the troposphere can also be a response to the extreme states of the polar vortex itself (e.g. Charlton et al., 2004; Sigmond et al., 2013; Tripathi et al., 2015b; Domeisen et al., 2020b). Forecast skill can be enhanced after stratospheric

extreme events (Sigmond et al., 2013; Tripathi et al., 2015b), but can also be reduced since the forecasts can be overconfident (Büeler et al., 2020; Statnaia and Karpechko, 2024), especially over Europe (Domeisen et al., 2020b). In particular, tropospheric internal variability can limit the coupling of stratospheric variability to the troposphere (Domeisen et al., 2020c). For instance, following the 2018 SSW event, the uncertainty in the development of synoptic activity after the SSW onset impacted the predictability of surface anomalies (González-Alemán et al., 2022).

Given that the uncertainty in the stratosphere is coupled to uncertainty in the troposphere, this study aims to systematically investigate the link between stratospheric and tropospheric uncertainty in the ECMWF subseasonal-to-seasonal (S2S) hindcasts and to identify regions and pathways for which better model representation might enhance the skill of stratospheric prediction.

## 2 Data and Methods

The Northern Hemispheric (NH) winter (November to February) subseasonal-to-seasonal (S2S) hindcasts (Vitart et al., 2017) of ECMWF model versions CY43R3 and CY45R1 are analyzed for the period 1998/99 to 2017/18. The hindcasts consist of 11 ensemble members, are integrated for 46 days and initialised twice a week. Both versions share similar configurations and are initialized with the ECMWF ERA-I reanalysis (Dee et al., 2011).

In addition, a hindcast for a case study initialized on 2018-01-27, 16 days before the onset of the 2018 SSW event on 2018-02-12, is chosen for a re-run to investigate the development of the large ensemble spread associated with this particular hindcast. This specific hindcast initialization date is chosen for a re-run because it displays a larger ensemble spread and consists of a larger portion of ensemble members that successfully predict the SSW event than the initializations available from the ECMWF real-time forecast on neighbouring dates (Figure A1). The hindcast is computed for an increased ensemble size (51 members compared to 11 in the original hindcast) and for more pressure output levels to enable a more robust investigation of the spread. The hindcast is re-run using model version CY47R3, computed on 2022-01-27, and is initialized with ERA5 reanalysis (Hersbach et al., 2020). The daily means of the 20-year hindcasts of the same model version are chosen as the climatology to compute anomalies for the hindcasts.

The zonal mean zonal wind at 60°N and 10 hPa ($U_{10,60}$) is used as a measure of the strength of the stratospheric polar vortex. As a measure of upward wave activity in the lower stratosphere, we use the zonal average of meridional eddy heat fluxes ($\overline{v'T'}$) averaged over 40-80° N at 100 hPa and weighted by the cosine of latitude, where $v$ is the meridional wind, $T$ is the temperature, and prime ($'$) denotes the departure from the zonal mean.

Hindcasts are categorized based on their ensemble spread in $U_{10,60}$. The uncertainty is estimated by first calculating the daily standard deviation of $U_{10,60}$ across the ensemble members of each hindcast. These daily standard deviations are then averaged over the 46-day integration period of the hindcast to obtain an estimate of the overall uncertainty present in the hindcast. Based on this 46-day average uncertainty, the hindcasts are separated into composites of large and small uncertainty, each consisting of 328 hindcasts. Specifically, the large uncertainty composite (large $U_{10,60}$ spread) is composed of hindcasts with an ensemble spread above the 75th percentile of all hindcasts (9.16 $ms^{-1}$), and the small uncertainty composite (small $U_{10,60}$ spread) is composed of hindcasts with an ensemble spread below the 25th percentile (5.86 $ms^{-1}$). Similar separations of hindcasts are

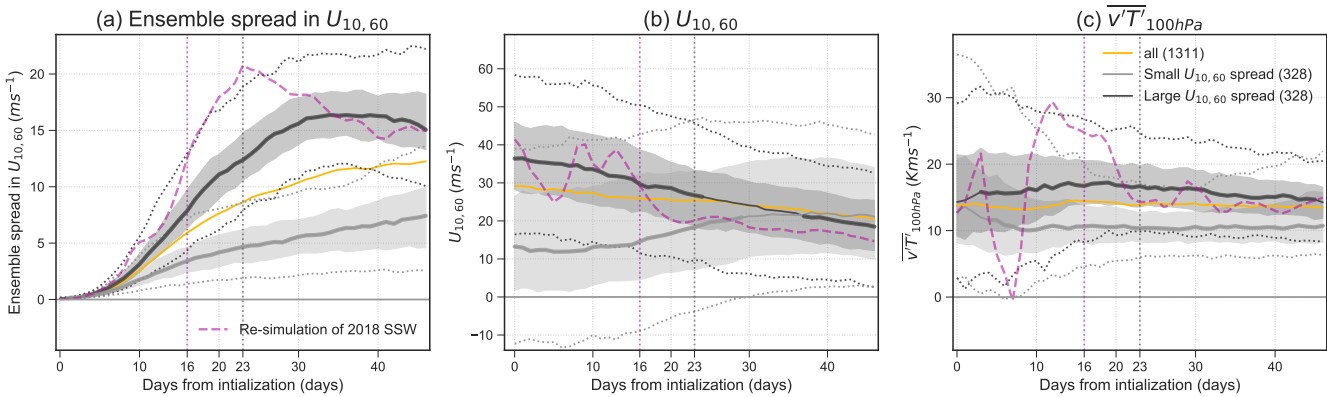

**Figure 1.** Evolution of (a) ensemble spread in $U_{10,60}$ and (b) ensemble mean of $U_{10,60}$ and (c) ensemble mean of $\overline{v'T'}$ at 100hPa in composites of hindcasts classified as having large uncertainty (large $U_{10,60}$ spread, black) and small uncertainty (small $U_{10,60}$ spread, grey), respectively, for the prediction of the stratospheric polar vortex. The solid line denotes the median, the shaded region denotes the 25th to 75th percentiles, and dotted lines denote the 5th and 95th percentiles, for the large and small spread composites. The median of all hindcasts is shown in yellow. Solid lines are printed in bold when the composites are significantly different from all hindcasts at the 95% confidence interval using a t-test. The purple dashed line in (a) corresponds to the ensemble spread of the hindcast of the 2018 SSW event, and the purple dashed lines in (b) and (c) correspond to the ensemble mean of the hindcast for the 2018 SSW. Dotted vertical purple and grey lines indicate the onset and the peak of the uncertainty in $U_{10,60}$ for the 2018 SSW event, respectively. The number of hindcasts in each composite is given in brackets in the legend.

found when using shorter averaging windows instead of the full 46-day average, and the results do not change significantly based on the definitions used (not shown).

## 3 Uncertainty in the Ensemble Prediction of the Stratosphere

We start by comparing and characterizing the features of high and low uncertainty hindcasts in the ECMWF subseasonal-to-
seasonal (S2S) model. Hindcasts that exhibit large uncertainty in the prediction of the strength of the stratospheric polar vortex ($U_{10,60}$) are associated with strong growth in the spread at around 5-25 days after initialization (Figure 1a). For hindcasts that exhibit small uncertainty, the spread in $U_{10,60}$ grows as lead time increases, but the rate of increase is much smaller than for the large uncertainty composite. Hereafter, the large uncertainty and small uncertainty composites are referred to as large $U_{10,60}$ spread and small $U_{10,60}$ spread composite, respectively.

The ensemble mean evolution in $U_{10,60}$ of the identified composites (Figure 1b) shows that on the day of initialization (day 0), the large $U_{10,60}$ spread hindcasts are more generally associated with a strong vortex and the small $U_{10,60}$ spread hindcasts are associated with a weak vortex, with the medians of the composites being $36.28 \ ms^{-1}$ and $13.25 \ ms^{-1}$ on day 0, respectively. After day 0, the large $U_{10,60}$ spread composite shows an overall weakening of the vortex and the small $U_{10,60}$ spread composite shows an overall strengthening of the vortex. The $U_{10,60}$ evolution of the composites is likely related to the

fact that SSWs or vortex weakenings in the large $U_{10,60}$ spread composite occur predominantly at relatively long lead times (from 10 days after initialization), while the SSWs or vortex weakenings in the small spread composite occur mostly at early lead times (within the first 10 days after initialization) (Figure A2). The difference in vortex strength between the composites reduces with lead time but remains significantly different from that of all hindcasts until 24 and 29 days after initialization, for the large and small $U_{10,60}$ spread composites, respectively. Towards longer lead times, from around 35 days after initialization,

the composites display a vortex strength similar to all hindcasts, likely linked to the model's drift towards climatology at long lead times. After that, the small $U_{10,60}$ spread composite stagnates at a vortex strength similar to all hindcasts, while the large $U_{10,60}$ spread composite weakens further and shows significantly weaker vortex strength than all hindcasts starting on day 37, possibly due to the stronger than average wave activity of the composite, which lasted until the end of the hindcasts (Figure 1c).

The respective behavior of the composites is consistent with our understanding that when the stratospheric mean flow is westerly, vertical wave propagation in the NH is possible for small wavenumbers (Charney and Drazin, 1961), while the exact propagation properties of the waves are modulated by the three-dimensional structure of the stratospheric flow. A strong vortex can further act as a waveguide, guiding waves from the troposphere towards the polar stratosphere (Matsuno, 1970; Simpson et al., 2009; Albers and Birner, 2014). On the other hand, when the vortex in the lower stratosphere is very weak, such as after

an SSW event, waves can be inhibited from propagating upwards, and the vortex can strengthen radiatively (Limpasuvan et al., 2005; Hitchcock and Shepherd, 2013). Indeed, as expected, the large $U_{10,60}$ spread composite that is associated with a stronger vortex is associated with stronger eddy heat flux in the lower stratosphere, as compared to the small $U_{10,60}$ spread composite, which is associated with a weaker vortex and weaker eddy heat flux (Figure 1b and c).

    To better understand the regional contributions to the spread in $U_{10,60}$, we now investigate the longitudinal structure of the

lower stratospheric heat flux (Figure 2). The large $U_{10,60}$ spread composite shows anomalously positive eddy heat flux over the North Pacific (NP), Northern Europe (NE), Siberia (Sib) and anomalously negative heat flux over North America (NA) (Figure 2a). The heat flux associated with NP peaks in the first few days after initialization, while that in the NA peaks after 10 days and in the NE after 15 days. For the small $U_{10,60}$ spread composite, the heat flux is weaker than for the large $U_{10,60}$ spread composite (Figure 2b) and comparable to the average of all hindcasts (yellow contours in Figure 2b). The heat flux of the

small $U_{10,60}$ spread composite is strongest at initialization and gradually decreases within the first 10 days for all longitudes. Interestingly, the heat flux over the North Pacific of the small $U_{10,60}$ spread composite increases again around 25 days after initialization, which might explain the stagnation of the increase in $U_{10,60}$ for the small spread composite in Figure 1b. The largest difference in the ensemble mean heat flux between the composites is found over the North Pacific owing to the very strong positive heat flux over the North Pacific associated with the large $U_{10,60}$ spread composite (Figure 2c).

In terms of ensemble spread, the large $U_{10,60}$ spread composite shows large uncertainty in the heat flux in all regions that also exhibit large positive and negative ensemble mean heat flux (Figure 2d). For the small $U_{10,60}$ composite, uncertainty is found in the same regions as for the large $U_{10,60}$ composite, but the ensemble spread is much weaker (Figure 2e). The largest difference between the high and low spread composites in descending order is over Northern Europe, followed by North America, the North Pacific, and Siberia (Figure 2f).

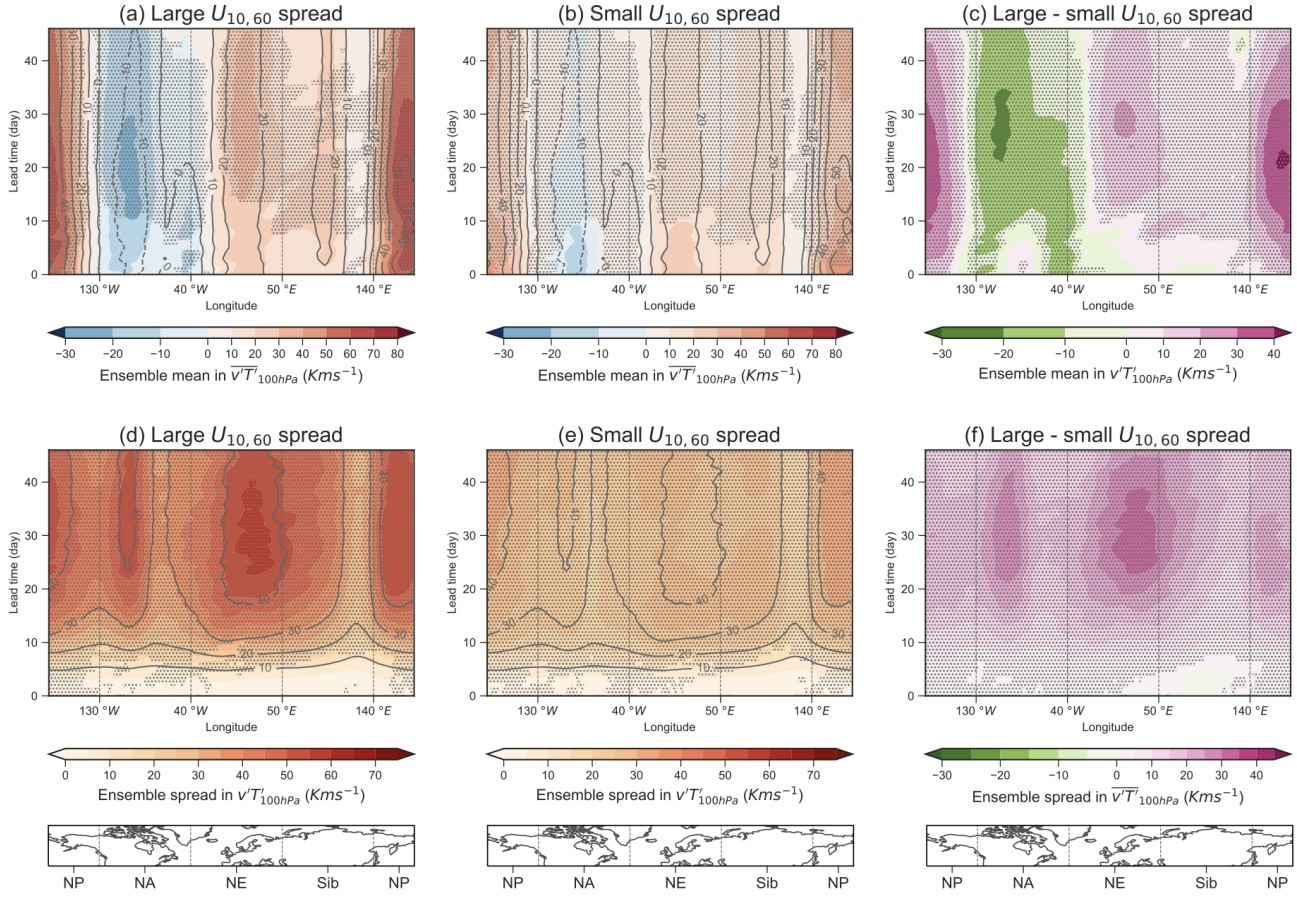

**Figure 2.** Hovmöller diagrams of composite ensemble mean and ensemble spread of $v'T'$ at 100hPa for (a,d) hindcasts with large spread in $U_{10,60}$ and (b,e) hindcasts with small spread in $U_{10,60}$. The difference between the composites, given as large minus small spread composite, in the ensemble mean and ensemble spread is displayed in (c) and (f), respectively. The averages over all hindcasts are plotted in grey contours. Stippling indicates significant differences at the 95% confidence level determined by a t-test in (a-b), (d-e) between the corresponding hindcast composite and all hindcasts and in (c), (f) between the hindcast composites. The grey vertical lines separate the regions of investigation, from left to right: North Pacific (NP, 140°E - 130°W), North America (NA, 130°W - 40°W), Northern Europe (NE, 40°W - 50°E) and Siberia (Sib, 50°E - 140°E). Note that the negative range of the colorbars is smaller than the positive range for visualisation purposes, but the contour levels are kept constant.

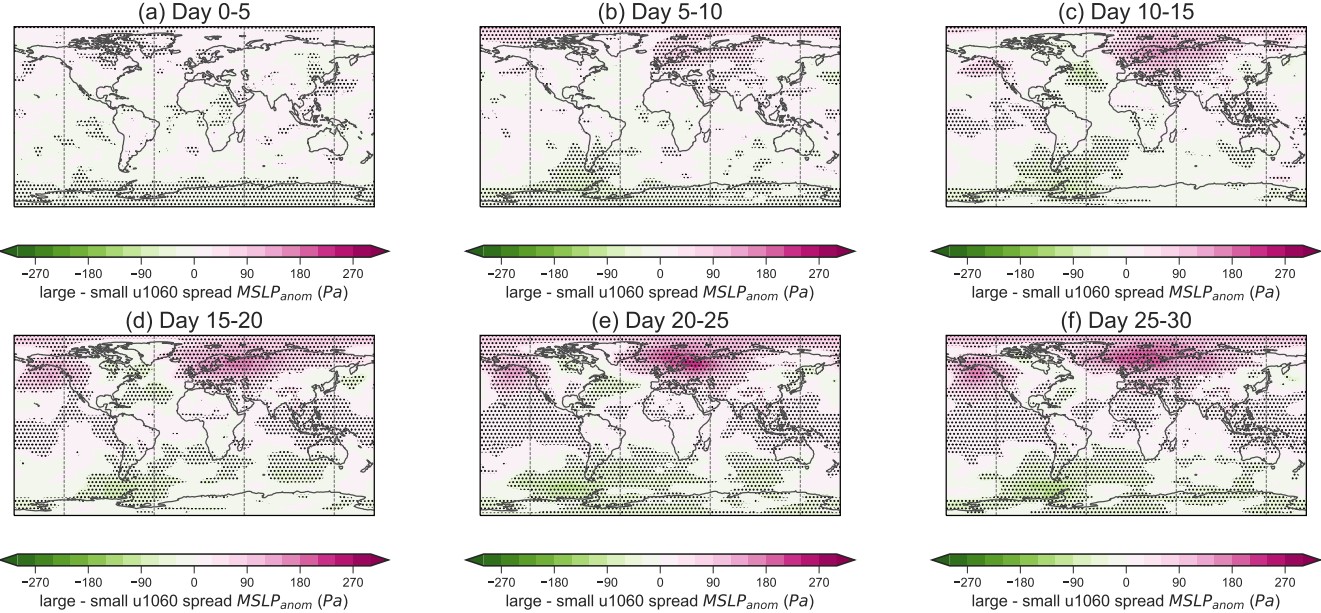

**Figure 3.** Difference in the evolution of composite ensemble spread of mean sea level pressure anomalies ($MSLP_{anom}$) given by hindcasts of large $U_{10,60}$ spread minus small $U_{10,60}$ spread. Differences that are significant at the 95% confidence level according to a t-test are marked by stippling.

## 4   Tropospheric Links to Stratospheric Uncertainty

As a next step, we investigate whether the uncertainty in the stratosphere is related to uncertainty in the troposphere by comparing the temporal and spatial evolution of the uncertainty of the large and small u1060 composites in mean sea level pressure (MSLP) anomalies (Figure 3). In the first 5 days after initialization, only small significant patches of larger uncertainty are found in the large $U_{10,60}$ spread composite compared to the small spread composite (Figure 3a). At days 5 - 10, a significant difference between the large and small $U_{10,60}$ spread composite is found over the North Pacific, the polar regions, Northern Europe and the Ural region. The difference in uncertainty between the composites at these regions persists and amplifies as lead time increases (Figure 3b - f), especially over the North Pacific and Scandinavia.

Other regions with significant differences between the large and small spread composites include the Azores High and the tropics during days 10 - 30 (Figure 3c - f). Smaller uncertainty is found in the large $U_{10,60}$ spread composite than the small $U_{10,60}$ spread composite over the Azores High during days 10 - 25 (Figure 3c - e). In the tropics, a small but significant difference is found from days 10 - 15 over the Maritime Continent and the tropical Pacific Ocean where the large $U_{10,60}$ spread composite shows larger uncertainty than the small $U_{10,60}$ composite (Figure 3c). The difference in uncertainty between the composites expands to more regions in the tropics and subtropics as lead time increases (Figure 3c - f), including Africa at around day 25 - 30 (Figure 3f). Small significant differences are also found in the Southern Hemisphere extratropics and

over Antarctica. These anomalies, especially at longer lead times (Figure 3d - f), may be connected to tropical precursors, such as for example the MJO (Stan et al., 2022) or El Niño Southern Oscillation (Taschetto et al., 2020). The phases of these phenomena are related to the strength of the Northern Hemisphere polar vortex and its predictability (Garfinkel and Schwartz, 2017; Domeisen et al., 2019), according to which the ensemble was separated here, but they also exhibit teleconnections to the Southern Hemisphere (e.g. Rondanelli et al., 2019; Taschetto et al., 2020).

The regions in the troposphere where uncertainty emerges are consistent with precursor regions that are known to modulate upward wave propagation into the stratosphere, namely over the North Pacific and Northern Europe (Garfinkel et al., 2010; Barriopedro and Calvo, 2014), and over Scandinavia and the Ural mountains, regions where increased blocking frequency occurs before SSWs (Martius et al., 2009; Peings, 2019). The consistency between the identified tropospheric origins of uncertainty and the precursor regions might suggest a propagation of uncertainty from the troposphere into the stratosphere through uncertainty in upward wave propagation, associated with uncertainty in tropospheric stationary waves (Schwartz et al., 2022) and in synoptic-scale conditions located in these regions (Lee et al., 2019, 2020). Larger uncertainty in the tropospheric stationary wave anomalies is associated with the large $U_{10,60}$ spread composite as compared to the small $U_{10,60}$ spread composite over the North Pacific, North America and Northern Europe at lead times beyond 20 days (Figure A3f). This uncertainty in the stationary waves might have contributed to the uncertainty in upward wave propagation (Figure 2f), as suggested in Schwartz et al. (2022). Tropospheric variability in these regions could thus contribute to the polar vortex weakening in the large $U_{10,60}$ spread composite, in which SSWs in the composite occur mainly at lead times of more than 10 days (Figure A2b).

At the same time, several of these regions are known to be impacted by stratospheric forcing, e.g. after SSW events. SSW can have downward impact over the Euro-Atlantic sector, resulting in a shift of storm track position (Afargan-Gerstman and Domeisen, 2020; Maycock et al., 2020), in a change of cyclone frequency (Afargan-Gerstman et al., 2024), and in the transition of weather regimes (Charlton-Perez et al., 2018; Domeisen et al., 2020c). Hence, since SSW events occur more frequently within the first 10 days after initialization in the small $U_{10,60}$ spread hindcasts (Figure A2c), the regions highlighted at longer lead times (Figure 3d - f) could also be related to downward impacts from the stratosphere. However, due to the substantial variability in the timing of SSW occurrence in both the large and small $U_{10,60}$ spread composites (Figure A2b, c), it is not possible to clearly determine whether these regions correspond directly to upward or downward coupling in these composites at a given lead time. Therefore, in Section 5, we further investigate the upward and downward pathways in a case study of the 2018 SSW prediction.

## 5 Development of the High Uncertainty in the 2018 SSW Prediction

A case with particularly high uncertainty in the prediction of the stratosphere was the SSW event on February 12, 2018. This case therefore represents a prime example for studying the origins of stratospheric uncertainty and their link to the troposphere. Furthermore, this event had a wide range of surface impacts (e.g. Kautz et al., 2020; Ayarzagüena et al., 2018; Hitchcock et al., 2022), while its prediction itself exhibited high uncertainty despite a range of suggested precursors, including extratropical

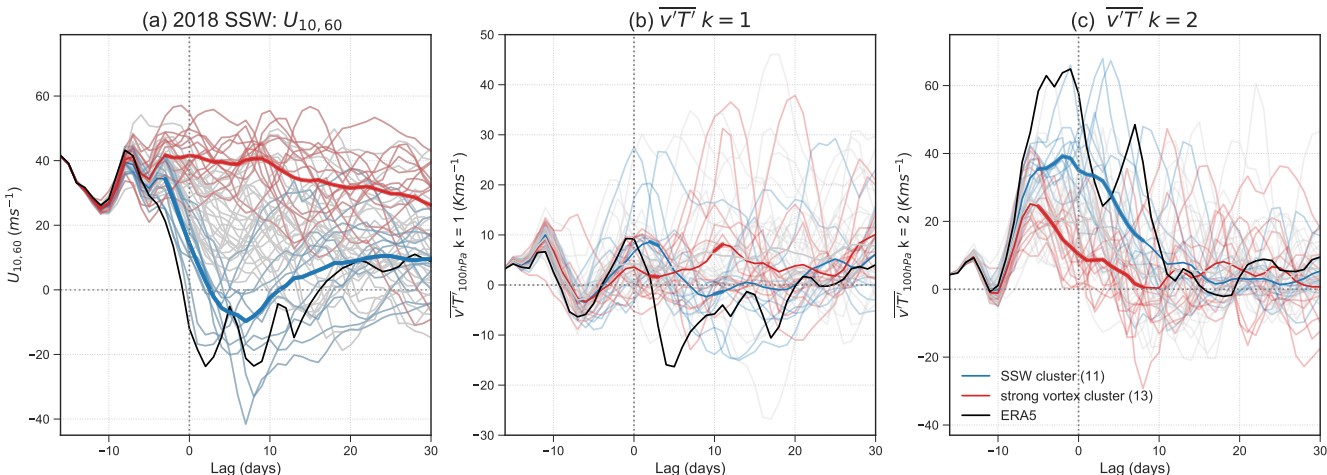

**Figure 4.** Ensemble plumes of (a) $U_{10,60}$ and $\overline{v'T'}$ at 100 hPa averaged over 45-75° N for (b) wave-1 and (c) wave-2, respectively, for the hindcast of the 2018 SSW event. Ensemble members are separated into strong vortex cluster (red) and SSW cluster (blue). The dark-colored solid lines denote the median of the composite. Solid lines are printed in bold when the ensemble clusters are significantly different from each other at the 95% confidence interval using a t-test. The black line denotes ERA5. The vertical line denotes the central date of the SSW on February 12, 2018.

troughs and blockings (Rao et al., 2018; Karpechko et al., 2018; Lee et al., 2019), and an MJO teleconnection (Statnaia et al., 2020).

We therefore further explore the development of uncertainty for the case study of the 2018 SSW. For this purpose we use an additional hindcast initialization with a larger number of ensemble members, initialized 16 days before the onset of the 2018 SSW event (see Methods). This initialization is selected because it includes ensemble members that successfully predict the onset of the SSW event and members that erroneously predict a strong vortex state around the time of the SSW onset, contributing to the large spread in ensemble for $U_{10,60}$. The selected initialization date shows a particularly extreme spread in $U_{10,60}$ compared to other initialization dates, with the spread increasing beyond the 95th percentile of the climatology and peaking at 7 days after the SSW onset (purple dashed line in Figure 1a). Consistent with the characteristics of the large uncertainty hindcasts discussed in Section 3, the hindcast is initialized under a strong vortex state (Figure 1b) and is associated with strong eddy heat flux around 10-20 days after initialization (Figure 1c), consistent with the occurrence of the SSW. Similar to methods used in e.g. Kautz et al. (2020) and Cho et al. (2023), we separate the ensemble into two clusters, one with ensemble members that successfully predict the SSW (*SSW cluster*) and one that predicts a strong vortex state (*strong vortex cluster*) (Fig. 4a), to investigate the differences between the clusters that subsequently lead to different predictions of the vortex strength.

Before the onset of the SSW, the clusters do not differ significantly in wave-1 heat flux in the lower stratosphere, whereas they do differ significantly in wave-2 at around lag -5 (Figure 4b and c). Both clusters show an initial increase in wave-2

activity, but the wave activity of the strong vortex cluster decreases shortly after the initial increase. The observed difference
between the two clusters in the wave-2 activity suggests that accurately predicting the wave-2 activity is crucial for successfully predicting the SSW, in agreement with previous studies (Karpechko et al., 2018; Rao et al., 2018; Lee et al., 2019; Statnaia et al., 2020). Although the SSW cluster on average still underestimates the wave activity as compared to reanalysis, and as a consequence the vortex deceleration, several individual ensemble members predict eddy heat fluxes comparable to reanalysis.

To further understand the origin of the difference between the clusters in wave-2 activity, we compare the differences between
210 the clusters in terms of their respective anomalies of outgoing longwave radiation (OLR) (Figure 5a - b) and of geopotential height anomalies (Figures A4 and A5) before SSW onset and of mean sea level pressure (MSLP) anomalies before and after SSW onset (Figure 5c - h). Before SSW onset, for lags -14 to -1, the SSW cluster shows more enhanced convection over the Maritime Continent and suppressed convection over parts of Africa and South America than the strong vortex cluster (Figure 5a - b). During lags -14 to -8, the SSW cluster also shows a stronger negative pressure anomaly over the Northwestern Pacific and
215 a stronger positive pressure anomaly over Northwestern America and the North Atlantic (Figure 5c). There is also a wavetrain pattern over the extratropics in the Southern Hemisphere that could be related to the enhanced convection over the tropics (e.g. Stan et al., 2022; Henderson et al., 2018). During lags -7 to -1, for the SSW cluster, the high pressure anomaly over Scandinavia amplifies and stronger negative pressure anomalies over the North Atlantic and Eastern Siberia are found (Figure 5d). This pressure dipole between Scandinavia and the North Atlantic is remarkably similar to the pattern that is identified by
220 Kent et al. (2023) to be crucial for successfully predicting the 2013 SSW, which was also preceded by strong wave-2 flux. The simultaneous increase in positive pressure anomaly over Scandinavia and Alaska, combined with the reduced negative pressure anomaly over Eastern Siberia project onto a climatological wave-2 pattern, which likely forced the upward wave-2 activity flux (Figure 4c and A6c,d) by amplifying the climatological stationary waves (Garfinkel et al., 2010).

The development of extratropical precursors to the SSW could potentially be linked to the enhanced convection over the
225 tropics, particularly the low pressure anomaly over the Northwestern Pacific during lags -14 to -8 (Figure 5c), which has been suggested to be associated with MJO phase 6/7 (Garfinkel et al., 2012, 2014; Liu et al., 2014; Schwartz and Garfinkel, 2017). A closer examination of the build-up of these anomalies indicates that the SSW cluster starts to show stronger convection over the Maritime Continent a few days after initialization (Figure A4a and A5a), followed by a trough over the Northwestern Pacific (Figure A4b and A5b), and a ridge over Alaska (Figure A4b and A5c). During lags -7 to -4, the ridge over Alaska develops
into anomalies that project onto the Pacific North American (PNA) pattern and form a wave train into Northern Europe (Figure A4c), potentially contributing to the formation of the trough over the North Atlantic and the ridge over Scandinavia (Figure A4d and A5d).

The higher pressure over Scandinavia and the lower pressure over the North Atlantic in the SSW cluster as compared to the strong vortex cluster before the SSW onset (Figure 5d) persist and strengthen further after SSW onset, while the high pressure
anomaly extends further towards Greenland and then spreads across the Arctic (Figure 5e-h). Starting at lag 7, the anomalies start resembling the negative phase of the North Atlantic Oscillation (NAO) (Figure 5f - h), consistent with the downward impact associated with the SSW event that is observed in reanalysis (Figure A8).

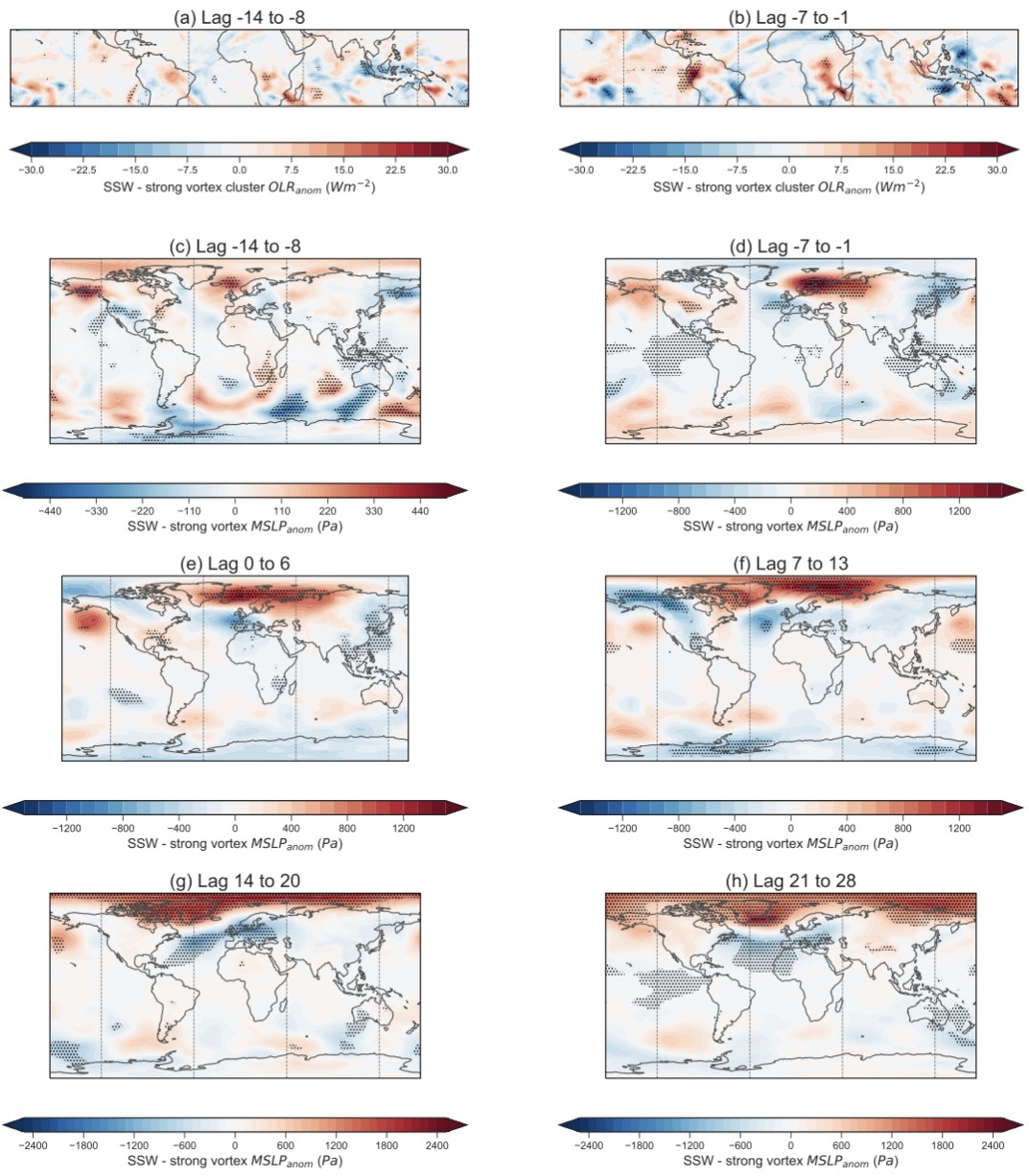

**Figure 5.** Difference between the SSW cluster and the strong vortex cluster in weekly averages of (a,b) outgoing longwave radiation (OLR) anomalies before SSW onset, and (c) - (h) mean sea level pressure (MSLP) anomalies before and after SSW onset for the hindcast of the 2018 SSW. Lag is given in days with respect to SSW onset. Anomalies are averaged every 7 days starting from 14 days before SSW onset (lag -14 corresponds to 2 days after initialization), for MSLP anomalies up to 28 days after SSW onset (lag 28). Stippling indicates a significant difference between the two clusters by a t-test at the 95% confidence level. Note that the upper and lower limits of the colorbars are increased from (c) to (h), with a colorbar range of $\pm 500Pa$ in (c), $\pm 1500Pa$ in (d) to (f), and $\pm 2500Pa$ in (g) and (h).

## 6 Conclusions

The uncertainty in the prediction of the stratosphere and the origins of the uncertainty are systematically investigated using
the S2S hindcasts of the ECMWF prediction system. By separating hindcasts into those that show large uncertainty versus
those that show small uncertainty in the prediction of the polar vortex strength ($U_{10,60}$), using ensemble spread as a measure of
uncertainty, we find that hindcasts associated with large uncertainty (large $U_{10,60}$ spread) tend to be initialized under a strong
vortex, while hindcasts associated with small uncertainty (small $U_{10,60}$ spread) tend to be initialized under a weak vortex.
Large $U_{10,60}$ spread hindcasts are also associated with a stronger ensemble mean wave activity in the lower stratosphere and
associated with larger uncertainty in the wave activity compared to small $U_{10,60}$ spread hindcasts. The characteristics of the
hindcast composites suggest that the vortex background state at initialization of a given hindcast can indicate whether the
uncertainty in the subsequent stratospheric prediction will be larger or smaller than average (compare also to Rupp et al., 2023;
Spaeth et al., 2024), and this relationship between hindcasts uncertainty and the vortex state can in turn be explained by the
different uncertainty in stratospheric wave activity under a different initial vortex state.

The difference in uncertainty between the hindcasts is further linked to the troposphere. Specifically, larger uncertainty
is identified over the North Pacific and Northern Europe in large $U_{10,60}$ spread hindcasts, where synoptic-scale variability
can modulate stratospheric vortex strength (Garfinkel et al., 2010; Martius et al., 2009) and impact the prediction of the
stratosphere (Kent et al., 2023). This tropospheric pattern suggests upward propagation of uncertainty from the troposphere
into the stratosphere through the uncertainty associated with the tropospheric stationary waves (Schwartz et al., 2022) and the
synoptic-scale conditions in these precursor regions (Lee et al., 2019; Karpechko et al., 2018). In turn, the stratosphere can
also propagate uncertainty downward, impacting predictability of the troposphere especially over the North Atlantic region
(Büeler et al., 2020; Spaeth et al., 2024; Statnaia and Karpechko, 2024). For instance, synoptic-scale tropospheric uncertainties
following stratospheric disruptions can limit the predictability of the troposphere (González-Alemán et al., 2022). Hence, the
identified uncertainty signal in the North Atlantic region is likely linked to both precursors and responses to stratospheric
extremes.

Since it is not possible to clearly separate tropospheric precursors and responses in the analysis of uncertainty for all cases,
as there are substantial overlaps of upward and downward coupling when considering all hindcasts together (not shown), the
upward and downward coupling of uncertainty between the troposphere and the stratosphere is further explored in a hindcast
of the 2018 SSW initialized 16 days before the event onset under a strong vortex. Initialized at the end of MJO phase 5 (Kiladis
et al., 2014) and near the onset of a record-breaking MJO phase 6 (Barrett, 2019), this event showed a particularly strong uncer-
tainty in the stratosphere ahead of the event onset. The hindcast's ensemble spans from erroneously predicting a strong vortex
to successfully predicting the observed SSW event. The ensemble members that successfully predict the SSW are preceded by
enhanced convection over the Maritime Continent and followed by a trough over the Northwestern Pacific, which is suggested
to be associated with MJO phase 7 (Garfinkel et al., 2014; Lin et al., 2017; Schwartz and Garfinkel, 2017). The development
of the trough over the Northwestern Pacific is followed by the development of a ridge over Alaska, a wave train to Northern
Europe, a trough over the Atlantic and a ridge over Scandinavia, and subsequently a development of wave-2 flux. Since the

ensemble members that successfully predict the SSW capture anomalies that are consistent with the extratropical impact of the MJO (Garfinkel et al., 2012, 2014; Liu et al., 2014; Schwartz and Garfinkel, 2017), and since the MJO is also suggested to act as a trigger for the SSW event (Statnaia et al., 2020), this hindcast of the 2018 SSW represents an example demon-

275 strating the propagation of uncertainty from the tropical troposphere into the stratosphere through teleconnection pathways (Schwartz and Garfinkel, 2017; Straus et al., 2023; Roberts et al., 2023). The ensemble members that successfully capture the MJO teleconnection and the SSW also better capture the downward impact associated with the SSW. Therefore, this hindcast also demonstrates the extended surface prediction skill that can be gained from the successful prediction of an SSW due to its precursors.

While tropospheric variability alone cannot fully explain the uncertainties in the stratosphere, and while not all wave activity that drives SSWs has a tropospheric origin (e.g. Birner and Albers, 2017), this study highlights how uncertainties in the troposphere can contribute to uncertainty in the stratosphere, and vice versa. Thus, a better representation of the regions identified in this study can be beneficial for both tropospheric and stratospheric prediction, in agreement with the suggested precursor regions of SSWs, e.g. over the North Pacific, the North Atlantic (e.g. Martius et al., 2009; Garfinkel et al., 2010),

and the tropics, for instance over the Maritime Continent for MJO teleconnections (e.g. Kang and Tziperman, 2018; Yadav et al., 2024). Model improvements for these regions, e.g. higher model resolution, improved representation of SST gradients and diabatic heating, may benefit the representation of the synoptic-scale conditions over the extratropics and, subsequently, the prediction of the stratosphere and its downward impacts.

**Appendix A**

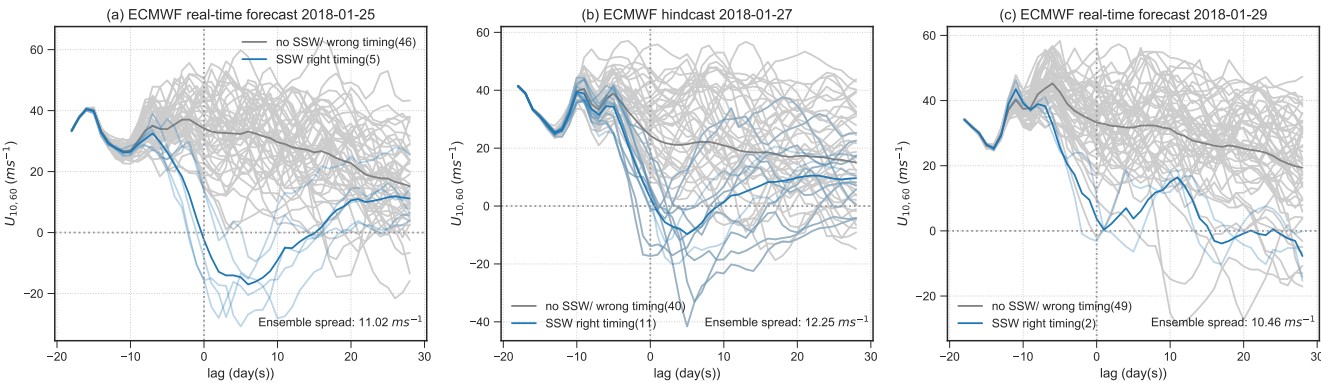

**Figure A1.** Ensemble plumes of zonal mean zonal wind at 10hPa, 60°N of ECMWF real-time forecasts initialized on (a) 2018-01-25 and (c) 2018-01-29, and (b) of the re-run of the ECMWF hindcast initialized on 2018-01-27. Blue lines denote members that successfully predict the 2018 SSW event within 10 days following the SSW onset and grey lines denote members that did not predict the SSW or that got the timing of the SSW wrong. Numbers in the brackets at the legend indicate the number of ensemble members in each category. Ensemble spread in zonal mean zonal wind at 10hPa, 60°N averaged over the entire hindcast / forecast period is indicated in the bottom right corner of each panel. Lag 0 denotes the onset of the 2018 SSW.

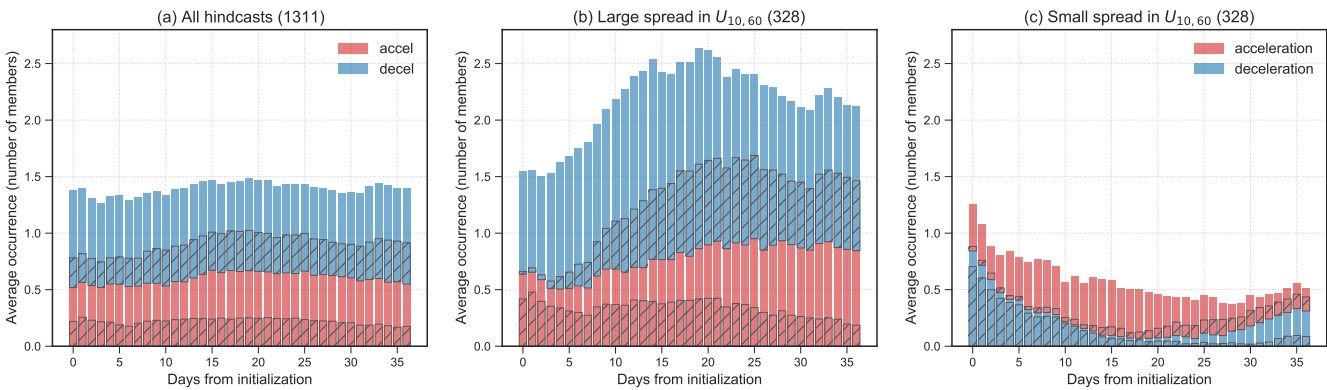

**Figure A2.** Average occurrence of 10-day strong wind acceleration and deceleration events, with event definitions following Wu et al. (2022), at a given day from initialization in (a) all hindcasts, (b) large $U_{10,60}$ spread hindcasts and (c) small $U_{10,60}$ spread hindcasts. Red and blue bars indicate the average occurrence of wind acceleration and deceleration events, respectively, in a 10-day window following a given day after initialization. Note that the blue bars and red bars are stacked on top of each other, and the bars together indicate the total average occurrence of wind acceleration and deceleration events in a given hindcast. The average number of events that evolve into an extreme state of the vortex, i.e. strong vortex events or sudden stratospheric warmings, during the 10-day event window are hatched.

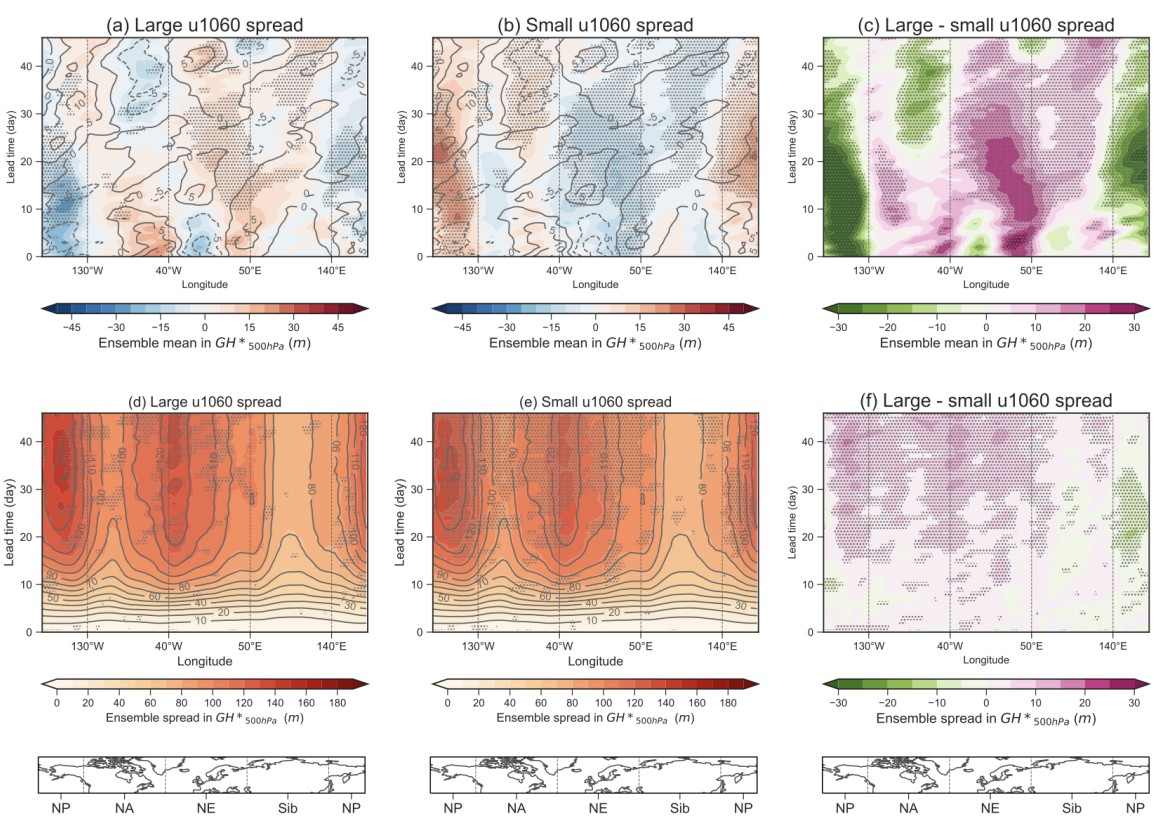

**Figure A3.** Same as Figure 2 but for zonal anomalies over geopotential height at 500 hPa ($GH*_{500hPa}$) averaged over 40-60°N.

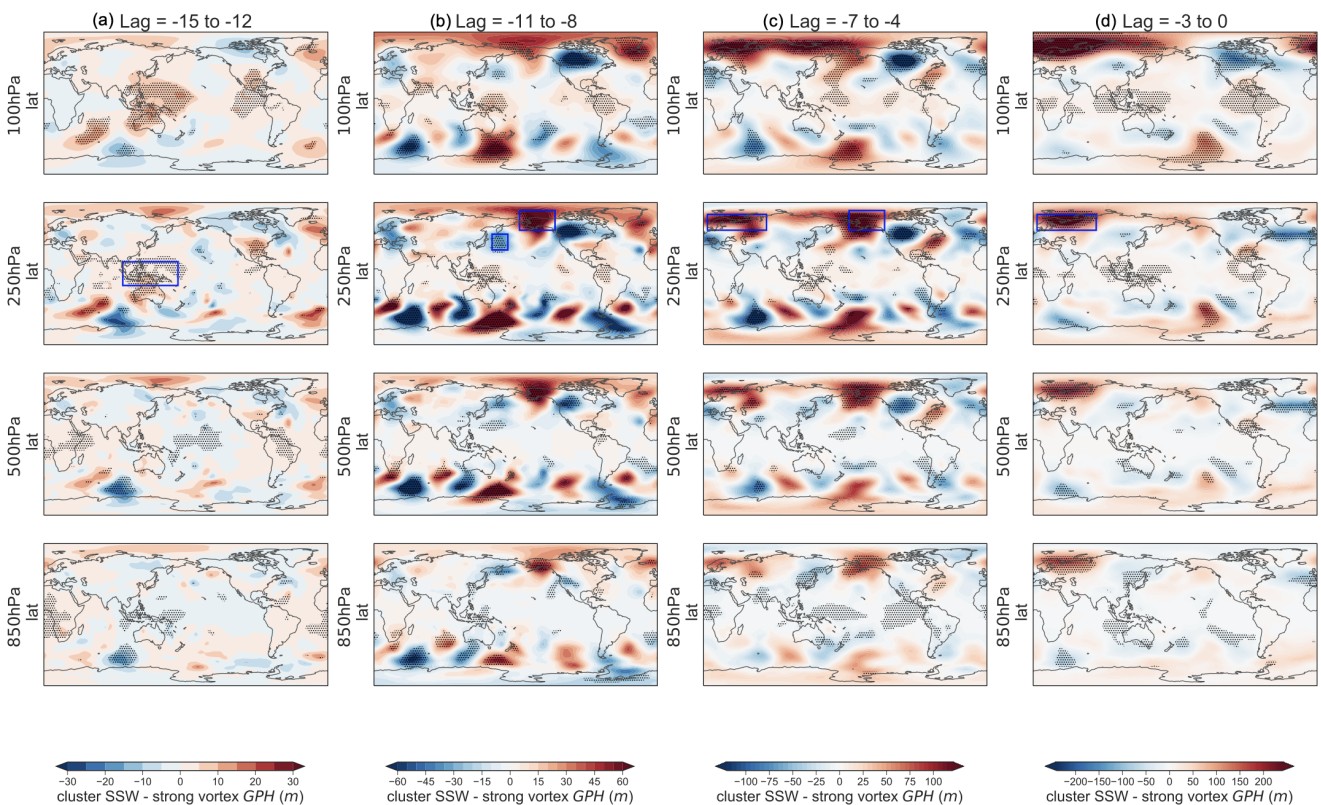

**Figure A4.** Difference between the SSW cluster and the strong vortex cluster in geopotential height at 100, 250, 500 and 850 hPa for the hindcast of the 2018 SSW, averaged every 4 days starting from 1 day after initialization (lag -15) to SSW onset (lag 0). Stippling indicates a significant difference between the two clusters by a t-test at the 95% confidence interval. Note that the range of the color bars is doubled with every time step from lag -15 to -12 to lag -3 to 0. Blue boxes in the 250hPa panels indicate regions where averages are taken for Fig. A5, from left to right, Maritime Continent (15°S-15°N , 100-170°E), Northwestern Pacific (30-50°N, 150-170°W), Alaska (55-80°N, 5-50°W) and Scandinavia (55-75°N, 5-80°E).

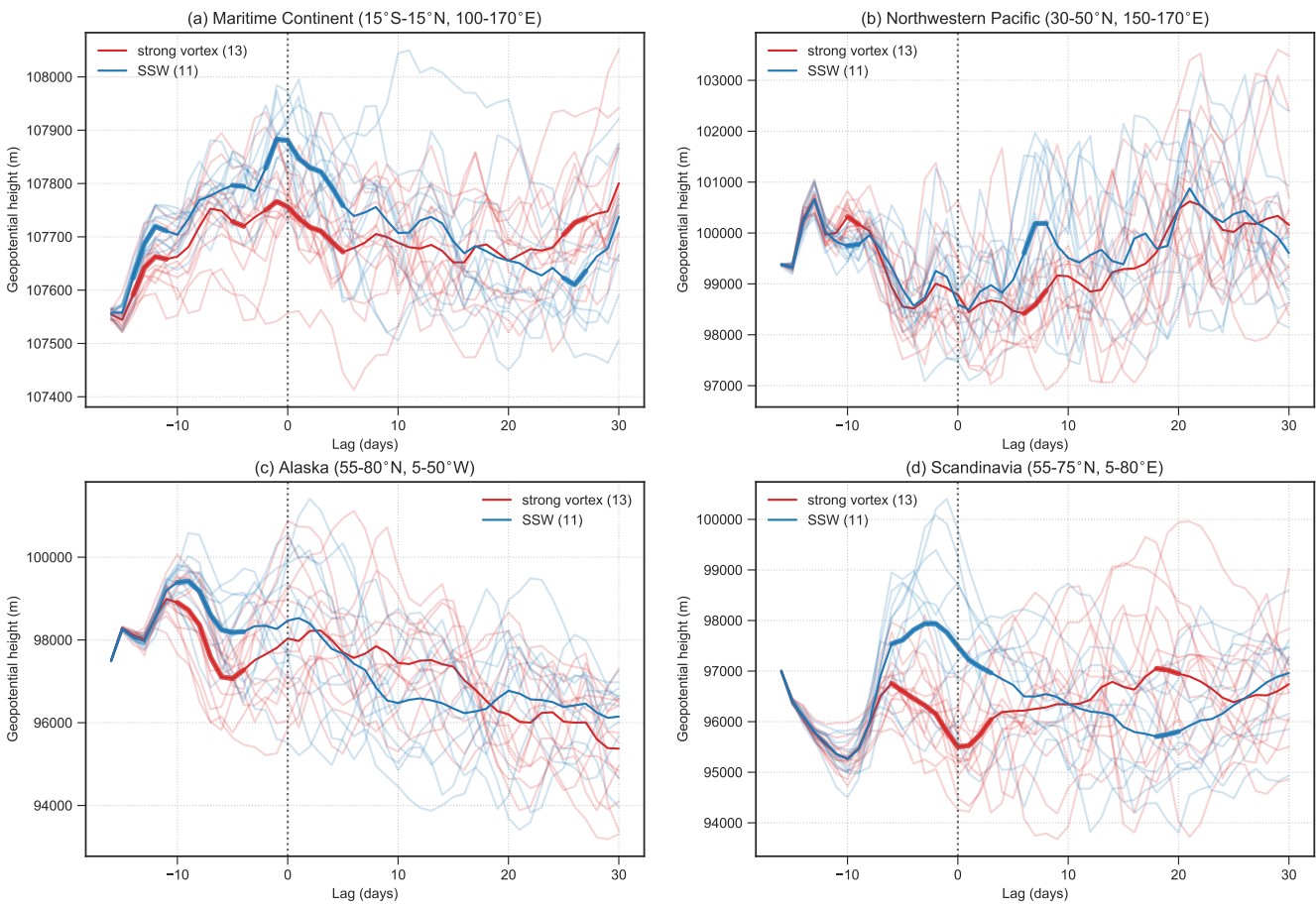

**Figure A5.** Ensemble plumes of geopotential height at 250hPa averaged over the following regions: (a) Maritime Continent (15°S-15°N, 100-170°E), (b) Northwestern Pacific (30-50°N, 150-170°W), (c) Alaska (55-80°N, 5-50°W) and (d) Scandinavia (55-75°N, 5-80°E). The regions are marked by blue boxes in Fig. A4. Ensemble members are separated into strong vortex cluster (red) and SSW cluster (blue). The dark-colored solid lines denote the median of the composite. Solid lines are printed in bold when the ensemble clusters are significantly different from each other at the 95% confidence interval using a t-test. The vertical line denotes the central date of the SSW on February 12, 2018.

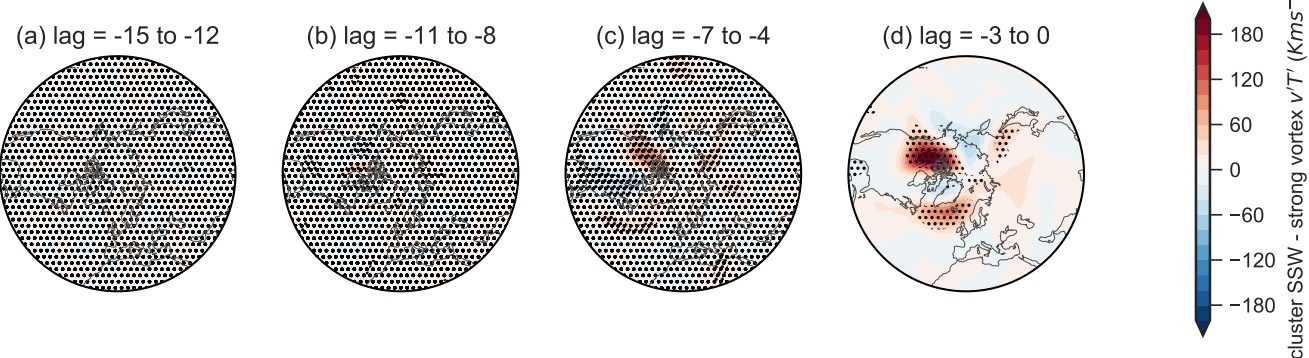

**Figure A6.** Difference between the SSW cluster and the strong vortex cluster in eddy heat flux ($v'T'$) at 100hPa for the hindcast of the 2018 SSW. Stippling indicates a significant difference between the two clusters by a t-test at the 95% confidence interval.

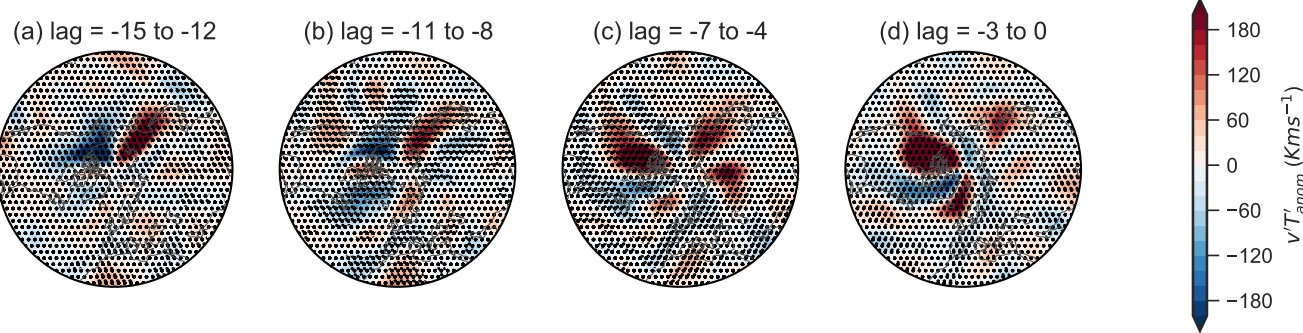

**Figure A7.** Eddy heat flux anomalies ($v'T'_{anom}$) at 100hPa before the onset of the 2018 SSW in ERA5 reanalysis. Stippling indicates a significant difference from climatology at the 95% confidence level.

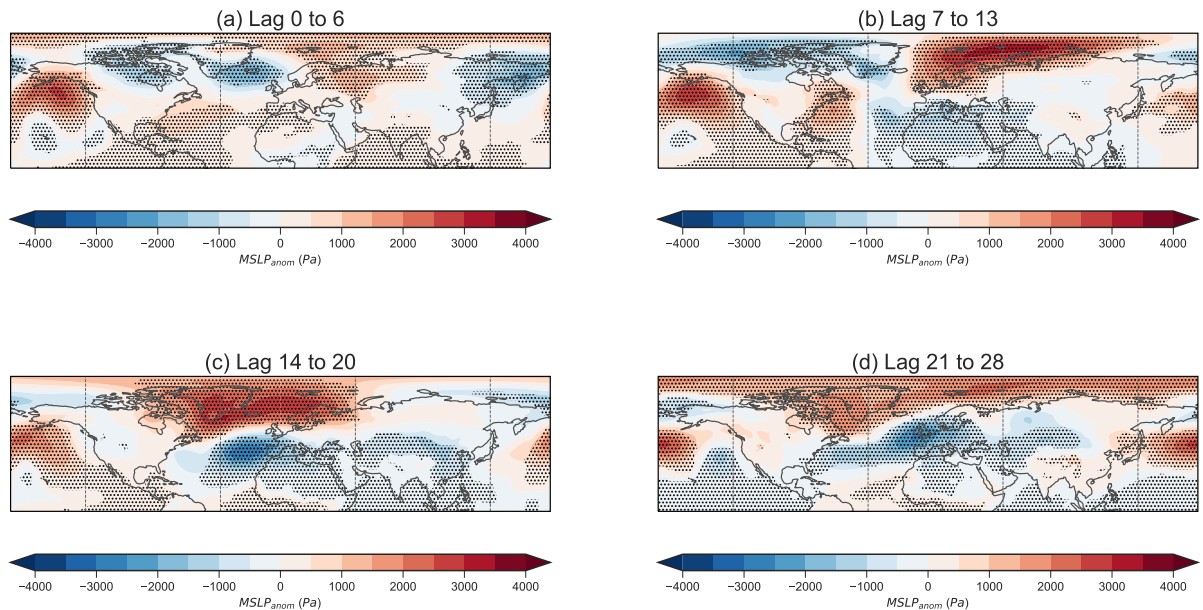

**Figure A8.** Mean sea level pressure anomalies after onset of the 2018 SSW in ERA5 reanalysis. Stippling indicates significant difference from climatology at the 95% confidence level.

*Data availability.* The ERA-Interim (Dee et al. (2011), https://cds.climate.copernicus.eu/#!/home; 2019) and ERA5 data (Hersbach et al. (2020), https://cds.climate.copernicus.eu/#!/home, 2023) are available from Copernicus Climate Change Service (C3S). The subseasonal-to-seasonal (S2S) data (Vitart et al. (2017), https://apps.ecmwf.int/datasets/data/s2s-reforecasts-instantaneous-accum-ecmf/levtype=sfc/type=cf/, 2019) is available from the ECMWF Public Dataset Service. The re-run of the hindcast data for the 2018 SSW event used in the study will be publicly available from https://doi.org/10.21957/hcmn-0572 (ECMWF (2024), https://doi.org/10.21957/hcmn-0572, 2024).

*Author contributions.* R.W. and D.D. designed the study. I.P. performed the re-run for the hindcast. R.W. performed the analysis, made the figures, and wrote the manuscript draft. R.W., D.D., G.C. and I.P. discussed the research and worked on revising the manuscript.

*Competing interests.* The authors declare no competing interests.

*Acknowledgements.* The authors would like to thank Frédéric Vitart, Andrew Charlton-Perez, Hilla Afargan-Gerstman and Zheng Wu for helpful discussions regarding this work. The work of R.W. is partly funded through ETH grant ETH-05 19-1 "How predictable are sud-

den stratospheric warming events?". Support from the Swiss National Science Foundation through project PP00P2_198896 to D.D., and PZ00P2_180043 to G.C. is gratefully acknowledged.

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
