# Peer review of "Tropospheric Links to Uncertainty in Stratospheric Subseasonal Predictions"

_EGUsphere, 2024_

## Referee Comment (RC3)

**Review: Tropospheric Links to Uncertainty in Stratospheric Subseasonal Predictions**

Rachel W.-Y. Wu, Gabriel Chioido, Inna Polichtchouk and Daniela I.V. Domeisen

July 02, 2024

**Summary**

This analysis identified and examined the uncertainty in the stratospheric predictions tracking its origins and attempting to establish a link to the subsequent tropospheric uncertainties. The authors constructed two groups of ECMWF hindcasts with large and small spread in the zonal mean zonal wind at 60N 10hPa averaged over the whole length of the forecast. The results of the group comparison show that the main regions of the uncertainty source in the stratosphere are associated with the precursor regions in the troposphere: North Pacific and Northern Europe. This suggests that the tropospheric uncertainty is linked to the stratospheric one. Moreover, the authors studied the SSW that took place in February 2018 in more detail to identify the tropospheric responses and establish the two-way propagation of uncertainty. The show that the ensemble members that predicted the SSW correctly also better predicted its downward impacts. In my opinion this is a very compact, yet somewhat conclusive study that proves existing hypotheses and provides new insights into the stratospheric-tropospheric coupling. Therefore, I recommend this manuscript for publication, but I also suggest a few comments below.

**Major comments**

- Averaging of the uncertainty over all 46-day integration period seems too broad and may filter out information. As the study focuses on the extended-range predictions it makes sense to average over week 3 and 4, for example. The authors may perform some sensitivity studies on that.
- The section which addresses the downward coupling and study of the SSW event seems to be not very well connected and lacking evidence. For example, the regions of the upward wave propagation could be discussed in this case. Also, as the authors argue that the SSW cluster better predicted the downward impacts, it would be helpful to add some comparison to reanalysis to prove that point.
- The OLR analysis does not seem to explain the difference between the clusters in wave-2 activity. I suggest the authors add some more explanation of the link to the tropics.

**Minor comments**

L34 I struggle to understand the part '..stationary waves, which interact with wave anomalies to determine the upward wave flux.' Please rewrite the sentence to make it clearer.

L48 Please consider adding '*often* resulting in better simulation of SSWs', as it is not always the case.

L52 Please consider adding (Statnaia and Karpechko, 2024) : a study investigating the overconfidence of the forecasts and its connection to the strength of the polar vortex.

L65 A new run for the 2018 SSW event is a very interesting addition to the study, but why you did not use the actual forecast issued on that date? Also, did you make only one run initialized on that date

or several to choose from? It is very interesting that quite a few ensemble members predicted the event well in advance.

L102 Do you have the explanation for the significant u1060 weakening after day 37 for the large u1060 spread composite?

Figure 1 Only the ensemble mean for the 2018 SSW events seems to be plotted (purple dash line). Please correct the figure caption.

References:

Statnaia, I. and Karpechko, A. Y. (2024) 'Factors influencing subseasonal predictability of northern Eurasian cold spells', *Quarterly Journal of the Royal Meteorological Society*, pp. 1–21. doi: 10.1002/qj.4744.

---

## Author Comment (AC1)

**Response to the reviewers of egusphere-2024-1652: Tropospheric Links to Uncertainty in Stratospheric Subseasonal Predictions**

Dear Editor Petr Šácha,

On behalf of all authors I would like to submit the revised version of the original article "Tropospheric Links to Uncertainty in Stratospheric Subseasonal Predictions" including an annotated version of the modified manuscript. During the revision, a number of changes have been made to the manuscript to satisfy the requests of the reviewers. Please find the summary of the changes and the detailed responses to the reviewers below.

Best regards,
Rachel Wu

We would like to thank all three reviewers for their helpful comments and suggestions for our study. These have been included into the manuscript (see changes indicated in **bold** in the annotated manuscript). Please find below the detailed responses (in blue) to the reviewers' comments and suggestions. All line indications refer to the new (annotated) version of the manuscript unless specified. The main changes to the manuscript are listed here:

**Changes to Figures:**

- Figure 2: Added stippling in the figure to indicate significance

- Added supplementary figures, Figure A1-A4, to Appendix

**Main topics of reviewer questions:**
Here, we list our answers to two common points raised by the reviewers. When these points are needed to be referred to in this document, they are referred to as *Main topics point 1* and *Main topics point 2*.

1 **Sensitivity test on the definition of hindcasts composites:** The method used to calculate the hindcast composites raised some concerns among the referees. We have therefore conducted sensitivity tests on Figures 1-3 (Fig. R1-R3). We have reproduced Figures 1-3 using different time averaging windows for the hindcast ensemble spread, i.e. using averaging windows of 0-22, 23-46 and 10-46 days after initialisation. The results of using 23-46 (Fig. R1g, R1m-r and R3m-r) and 10-46 day averages (Fig. R1a, R1a-f and R3a-f) give similar results as to taking an entire 46-day average as in the original manuscript (Figure 1-3).

For the 0-22 day average, by definition, the hindcast composites differ from other definitions in that the largest difference in spread between the composites is found in the first 23 days (Fig. R1d), where the other definitions, i.e. 10-46 and 23-46 days (Fig. R1a and g), have the largest difference in spread in the second half of the 46 days. The 0-22 day average also differs from other definitions in terms of the timing of vortex evolution of weakening and strengthening (Fig. R1e) and in terms of the timing in the difference in wave activity between the composites (Fig. R1f). The largest difference between the hindcast composites in wave activity for the 0-22 day average is found earlier (Fig. R1f and Fig. R2i), in the first 10 days, compared to other definitions in which the largest difference in wave activity is found around at least 10 days after initialisation (Fig. R1c,i and Fig. R2c,o).

Although the different definitions separate hindcasts into composites that display large ensemble spread differences at different time periods, these composites produce the same results. Independent of the definitions to separate hindcast composites, large uncertainty composites are associated with a strong vortex at initialisation, and small uncertainty composites are associated with a weak vortex (Fig. R1b,e,h). In periods where a large difference in ensemble spread is found between the hindcast composites (Fig. R1a,d,g), a large difference in wave activity (Fig. R2c,i,o) and in the spread of wave activity is also found (Fig. R2f,l,r). The robustness of the results suggests that the initial background state of the stratosphere can indicate whether the ensemble spread of a given hindcast will be larger or smaller than

average, and the timing and magnitude of wave activity can determine the timing and how large the ensemble spread will be.

In terms of the ensemble spread difference between the hindcast composites in mean sea level pressure (MSLP) anomalies (Figure 3), the 10-46 and 23-46 day definitions (Fig. R3a-f and m-r) also show a similar pattern as using the full 46-day definition in Figure 3. The 0-22 day definition also shares a similar pattern as all the other definitions, but the magnitude of the pattern is weaker at long lead times (Fig. R3g-l) compared to other definitions. Though the magnitude is not exactly the same across the definitions, the overall regions and pattern where the ensemble spread differs between the hindcast composites in MSLP anomalies are qualitatively the same. The results in Figure 3 are thus also robust across different definitions.

Since the definition of using a 46-day ensemble spread average separates hindcasts into composites that show large differences in the ensemble spread at long (subseasonal) lead times, we decided to keep the original definition for hindcast separation. From the sensitivity tests, we conclude that the results that large u1060 spread hindcasts are associated with a strong vortex at initialisation and that the small u1060 spread hindcasts are associated with weak vortex at initialisation are largely insensitive to the definition used. The time period of the largest difference in ensemble spread between the composites is linked to the time period when the difference in wave activity between the composites is the largest. The regions of the largest difference in ensemble spread between the hindcasts composite in MSLP anomalies are also robust across definitions. We have added a sentence at Lines 91-93 commenting on how the results are insensitive to the time-averaging window used to separate hindcast composites.

2 **Relevance of tropical convection to the development of wave-2 for the 2018 SSW:** Thank you for raising this point, and we agree that the relevance of tropical convection to the development of wave 2 was not sufficiently clear in the manuscript. We have now added supplementary figures to support the suggestion linking tropical and extratropical precursors to the development of wave-2. In Figure R4, we show the geopotential height anomalies at multiple pressure levels in the troposphere and in the lower stratosphere using shorter time averages (4-day averages), instead of the weekly averages shown in Figure 5, to allow for a better investigation of the development of the mid-latitude precursors from tropical precursors. Figure R4 is centered around the Pacific, so the tropical-extratropical connection is easier to observe. We also plot ensemble plumes of geopotential height at 250hPa averaged across 4 regions (Fig. R5), regions that are suggested in literatures to be related to tropical-extratropical teleconnection, to better quantify when and where the differences in anomalies between the two ensemble clusters emerge. This plot of the ensemble plume allows for a direct comparison to Figure 4.

The ensemble clusters that successfully predicted the SSW and those that predicted a strong vortex differ significantly from each other in terms of wave-2 starting at lag -5 and in terms of vortex strength (u1060) starting at lag -3 (Figure 4a and 4c). While before the significant difference in wave-2 between the two clusters, the ensemble clusters differ from each other within a few days after initialization in mean sea level pressure (Figure 5) and in geopotential height (Fig. R4). The SSW cluster is first associated with stronger convection over the Maritime Continent during lags -15 to -12 (Fig. R4a and Fig. R5a). During lags -11 to -8, a trough over the Northwestern Pacific (Fig. R4b and Fig. R5b) and a ridge over Alaska are found for the SSW cluster (Fig. R4b and Fig. R5c). At lags -7 to -4, the ridge over Alaska develops downstream into a Pacific-North American (PNA) pattern and a wavetrain to Northern Europe. A high pressure over Scandinavia and a trough over the North Atlantic develop at lags -7 and -4 (Fig. R4c) and persist until SSW onset (Fig. R4d and Fig. R5d). This pattern over the North Atlantic and Scandinavia during lag -7 to 0 (Fig. R4c and R4d) is believed to be crucial to the wave-2 development at lag -5 (Figure 4c) (Kent et al., 2023).

The trough that develops over the Northwestern Pacific is likely related to the tropical convection over the Maritime Continent, since the trough is consistent with the anomalies that are suggested to be associated with tropical convection and MJO phase 7 (Garfinkel et al., 2014, 2012; Liu et al., 2014). This trough over the Northwestern Pacific can contribute to upward wave propagation, especially for wave-1 Garfinkel et al. (2014), and as shown in this case study of the 2018 SSW, might also be related to the development of extratropical anomalies that may have an influence further downstream towards Northern Europe, and subsequently the development of wave-2.

[Figure]

Figure R1: Figure 1 re-plotted using different definitions to separate hindcast composites. Hindcast composites are separated using averaged ensemble spread during (a-c) 10-46 days, (d-f) 0-22 days and (g-i) 23-46 days after initialization.

[Figure]

Figure R2: Figure 2 re-plotted using different definitions to separate hindcast composites. Hindcast composites are separated using averaged ensemble spread during (a-f) 10-46 days, (g-l) 0-22 days and (m-r) 23-46 days after initialization.

[Figure]

Figure R3: Figure 3 re-plotted using different definitions to separate hindcast composites. Hindcast composites are separated using averaged ensemble spread during (a-f) 10-46 days, (g-l) 0-22 days and (m-r) 23-46 days after initialization.

[Figure]

Figure R4: Geopotential height difference between the SSW cluster and strong vortex clusters, averaged every 4 days starting from 1 day after initialisation (lag -15) to onset (lag 0). Stippling indicates significant difference between the two clusters using a t-test at the 95% confidence level. Note that the range of the colorbars doubles from one panel to the next, from (a) to (d). Blue boxes at the 250hPa panels indicate the regions where averages are taken for Fig. R5, from left to right, Maritime Continent (15°S-15°N , 100-170°E), Alaska (55-80°N, 5-50°W) and Scandinavia (55-75°N, 5-80°E).

Figure R4 and R5 are now included in the Appendix as Figures A2 and A3. We have modified the text to better illustrate the link of tropical and extratropical precursors to the development of wave-2 (Lines 214-222). We have also modified the conclusions accordingly in Lines 251-266.

**Reviewer 1**

**General Comments**

The paper is well written and relevant for the community. Its aim is to evidence the link between tropospheric and stratospheric uncertainty in a climatological context, using subseasonal ensemble hindcasts issued by ECMWF. The focus on the example of the 2018 SSW prediction is useful to make a link with individual subseasonal forecast cases. After the comments below have been addressed, I will be happy to advise pubblication of the manuscript in Weather and Climate Dynamics.

**Specific Comments**

Lines 23-24 : From this sentence it seems as though the only sources of uncertainty for the stratospheric variability are the mean state of the stratosphere and the vertical wave propagation from the troposphere. Could you make clear that vortex preconditioning, also regulated by internal stratospheric oscillations, is an important factor modulating vertical wave propagation from the troposphere (hence stratospheric variability) ? For discussing this, I would advise to check the references in the second and third paragraph of the introduction of De la Cámara et al. 2019.

Thanks for this comment. I have gone through the second and third paragraph of the introduction of De la Cámara et al. 2019 and included some of the references from there, changes are made in Lines 23-34.

[Figure]

Figure R5: Ensemble plumes of geopotential height at 250hPa averaged over regions (a) Maritime Continent (15°S-15°N, 100-170°E), (b) Northwestern Pacific (30-50°N, 150-170°W), (c) Alaska (55-80°N, 5-50°W) and (d) Scandinavia (55-75°N, 5-80°E). The regions are marked by blue boxes in Fig. R4. Ensemble members are separated into a strong vortex cluster (red) and an SSW cluster (blue). The dark-colored solid lines denote the median of the composite and solid lines are printed in bold when the ensemble clusters are significantly different from each other at the 95% confidence interval using a t-test. The vertical line denotes the central date of the SSW on February 12, 2018.

Line 78 : Your measure of (average) spread, used to separate hindcasts in large and small uncertainty groups, mixes short-term and subseasonal time ranges. Would the hindcast selection change if the spread was averaged starting from time +1 week (or +10 days) until the end of the hindcast period? Can you re-compute Figure 1 with this new selection as a sensitivity test ?

*Thanks, we have conducted the sensitivity test as mentioned above in Main topics point 1. Similar hindcast selection is found by using the full 46-day period of the hindcast or a window of 10 to 46 days or 23 to 46 days, which all separate hindcasts into composites which show large ensemble spread at long lead times. As described in point 1, the conclusion that large ensemble spread composite is associated with strong initial vortex state and small ensemble spread composite is associated with weak initial vortex state is insensitive to the definitions used. We added a sentence to comment on the robustness of the results to hindcast composite definitions in Lines 91-93.*

Figure 2 : It would be useful to show significance from the hindcast climatology (in a,b,c,d) and between the two hindcasts groups (in c,f).

*Thanks a lot for the suggestion, we have now updated Figure 2 with stippling to indicate significance.*

Lines 218-232 : While the effect of mid-latitude precursors on the wave-2 propagation and the SSW development is quite clear and interpretable, the impact of the tropics is more subtle, and is suggested with a certain caution also in Statnaia et al. 2020. In Figure 5 a,b you show the tropical OLR before the event. How do you connect this directly to the anomalous propagation of wave-2 ? Does the mean error of the two ensemble means from the reanalysis in terms of OLR provide evidence that the tropics are better captured by the SSW ensemble ? In my opinion, in the Conclusions you stress a lot the impacts of the Tropics although the evidence supporting this is rather weak. The same also applies to a sentence in the Abstract.

*Thanks for the comment. As described in Main topics point 2, we have added supplementary figures (Figure R4 and R5 here in this document, Figure A2 and A3 in the manuscript) and more description on the link of tropical convection to the wave-2 development in the main text in Lines 214-227. We hope the supplementary figures and the added explanation will help to better support the link between the successful capturing of the enhanced tropical convection and the successful prediction of the 2018 SSW.*

Line 207 : It would be interesting to mention the reason for the large-spread ensamble having a stronger vortex at initial times.

*Thank you and we have added some more interpretations in Lines 235-239.*

**Technical corrections**

Line 7 : missing verb.

*Thank you, we have added a verb in line 7.*

Lines 18-22 : Break up in two sentences.

*Thank you, we have split the sentence in two at line 20.*

Lines 30-31 : Check incorrect wording.

*Thank you, we have modified the sentence, now on Line 35.*

Lines 37-41 : Break up in two sentences.

*Thank you, the sentence is split in two, now at Line 44.*

Line 48 : « To » or « with » instead of « into ».

Thank you, we have replaced "into" by "to", now at Line 54.

Lines 69-70 : Why do you mention the computation date ? I find this quite confusing when talking about hindcasts for a different date in 2018.

Thanks and we have left out the computation date and just to mention that hindcasts from the same model version are used, at Line 79.

Line 72 : I would suggest to change u1060 to something more clear like $U_{10hPa}{}^{60N}$

Thank you, we have changed u1060 to $U_{10,60}$.

Line 124 : Uncertainty is large also where the ensemble mean flux is anomalously negative.

Thanks and have modified at Line 137.

Line 240 : Repetition of « in this region ».

Thanks and have removed the phrase, now at Line 277.

Cámara, A. d. l., T. Birner, and J. R. Albers, 2019: Are Sudden Stratospheric Warmings Preceded by Anomalous Tropospheric Wave Activity?. J. Climate, 32, 7173–7189, https://doi.org/10.1175/JCLI-D-19-0269.1.

**Reviewer 2**

review of "Tropospheric Links to Uncertainty in Stratospheric Subseasonal Predictions" by Wu et al

This paper demonstrates that uncertainty in stratospheric S2S prediction originates in uncertainty in the troposphere. Using a large set of hindcasts from the IFS, they find that weak vortex cases tend to have more certainty as compared to strong vortex cases in the first few weeks. Also, large uncertainty cases tend to have larger mean values of 100hPa heat flux. They then perform a detailed investigation of the 2018 SSW, and show that intermodel spread in its prediction is specifically associated with intermodel spread in wave-2 in the troposphere.

The paper is already in good shape, however I think the arguments could be made stronger. Please see my suggestions below

**General Comments**

1. Is it possible to produce Figure 2 for a tropospheric level? The authors are connecting v'T' at 100hPa with the underlying tropospheric stationary waves. While this is, to first order, an ok assumption to make, the connection would be more immediate if the level shown was lower down. This is because the tropospheric waves can sometimes have zonal propagation and strong phase tilt near the tropopause, and hence the connection with the longitude of the underlying wave source can sometimes be lost.

   Thanks a lot for the suggestion. We have reproduced Figure 2 for 500hPa (Fig. R6). Large spread is found in both large u1060 spread and small u1060 spread composites at 500 hPa (Fig. R6d-e). The spread associated with large u1060 spread composite is also larger than the small u1060 spread composite at 500 hPa (Fig. R6f) as at 100 hPa, but the difference at 500 hPa is much smaller than the difference between the composites at 100 hPa (Figure 2f). The ensemble mean of the heat flux at 500 hPa is larger for the large u1060 spread composite than for the small u1060 spread composite at around 130°W, 40°W and 140°E but smaller in other regions (Fig. R6c). However, since the the differences between the hindcast composites with model climatology and the difference between the two hindcast composites are relatively small, Fig. R6 is not able to clearly connect v'T' at 100hPa (Figure 2) to tropospheric conditions.

   To better illustrate the connection with tropospheric stationary waves, we also plot the wave anomalies at 500hPa (zonal anomalies of 500hPa geopotential height) (Fig. R7). A significant difference is found

[Figure]

Figure R6: Same as Figure 2 but at 500 hPa.

[Figure]

Figure R7: Same as Figure 2 but for zonal anomalies over geopotential height at 500 hPa ($GH*_{500hPa}$) averaged over 40-60°N.

between the large and small u1060 spread composite. The large composite shows negative anomalies over the North Pacific and positive anomalies over Northern Europe, and vice versa for the small u1060 composite. The large u1060 spread composite is associated with anomalies favouring upward wave propagation, i.e. deepened Aleutian Low and higher pressure over Scandinavia (Fig. R7a). The small u1060 spread composite is associated with anomalies favouring suppressed wave activity, i.e. weakened Aleutian Low and lower pressure over Scandinavia (Fig. R7b). The difference between the composites in the wave anomalies at 500hPa is consistent with the stronger wave activity in large u1060 spread composite than the small u1060 composite (Figure 1c and 2c). The large u1060 spread composite also shows larger uncertainty in tropospheric stationary wave anomalies from around 15 days after initialisation over North Pacific and Northern Europe (Fig. R7f), in which these uncertainty in the tropospheric stationary waves can propagate into wave activity in the lower stratosphere (Figure 2f) (Schwartz et al., 2022).

We have now added Fig. R7 in the Appendix as Figure A1 and added Lines 161-166 to better illustrate the link between upward wave propagation and tropospheric stationary waves.

2. The authors define the high uncertainty composite and low uncertainty composite by averaging over the entire 46 days of the hindcasts. I wonder if the message regarding the direction of propagation of uncertainty would be clearer if the averaging was performed only over certain days. Specifically, if the averaging was over the second half only, the upward propagation of uncertainty could be better constrained by focusing on tropospheric and lower stratospheric conditions before. Similarly, if the averaging was over the first half only, the downward propagation of uncertainty could be better constrained by examining tropospheric conditions afterwards. The reason I suggest this is that the current Figure 3 and its accompanying text can be hard to interpret, as it isn't clear the direction in which uncertainty is propagating. (It could be that there is substantial overlap as to which hindcasts have uncertainty in the first vs. second half, in which case this suggestion won't be very helpful)

Thanks for the suggestion. The separation of hindcasts by taking the first half (0-22 days) and the second half (23-46 days) of the ensemble spread does separate hindcasts into composites that show larger difference in uncertainty and wave activity into the first half or the second half (Fig. R1a and R1c). However, these composites cannot be separated clearly into upward or downward coupling effects since the exact timing of upward or downward coupling can still be quite different between the hindcasts and within a given composite. The different definitions, therefore, also give qualitatively similar results when comparing the ensemble spread in MSLP in Fig. R3g-l and R3g-l that the large u1060 spread composite has a larger spread than the small u1060 spread composite over the North Pacific and Scandinavia. Therefore, we conclude that we are not able to clearly separate the effects of upward and downward coupling using the hindcasts in our study, and the direction of coupling is only investigated further in the 2018 SSW case study.

**Minor Comments**

section 5: Cho et al 2023 perform a very similar exercise but with a focus on the 2021 SSW. They find that for this event, transmission in the lowermost stratosphere is crucial. It would be helpful if the authors could try to diagnose the effect Cho et al found, so as to explore its relevance for the 2018 SSW. Even if this is not practical, this paper should be cited near e.g., line 37 and 41 among other places, and also included in the discussion. Furthermore, there are two recent papers that should also be discussed in the discussion: Spaeth et al 2024 and Rupp et al 2023. These papers are somewhat less directly related to this paper than Cho et al., but certainly the discussion section should place the present results in the context of this work

Thanks for the comment. At earlier stages of this study, we have also explored the effect of the vortex mean state and vortex structure in successfully predicting the 2018 SSW. However, as also shown in Figure 4a, the difference between the ensemble clusters that successfully predict the SSW and those that predict a strong vortex only differ in the vortex mean state around 3 days before SSW onset (Figure 4a). The clusters also differ in part of the vortex structure and meridional PV gradient around 5-6 days before the SSW onset (not shown). We therefore argue that the development of wave-2 from tropical and extratropical precursors plays a more

crucial role than the vortex background state in successfully predicting the 2018 SSW event. Cho et al 2023 is nevertheless relevant in the Introduction and is included in Line 40. Spaeth et al 2024 and Rupp et al 2023 are also relevant papers to include and they are included at Lines 237-238.

line 48: include Garfinkel and Schwartz 2017 here too.

Thanks, have included Garfinkel and Schwartz 2017 and now the sentence is at Line 54.

figure 1 caption: I don't see any indication of the ensemble spread for the 2018 case in the panels. Please remove this from the caption, or otherwise add it to the figures.

The ensemble spread is in Figure 1a, the purple horizontal dashed lines. We have modified the caption to make it more clear.

line 185, 226/227: are the convection anomalies this similar to the MJO event that was happening at the time? That is, are the events which capture the SSW also those which better simulate the MJO? See Garfinkel and Schwartz 2017 who argue that models which better capture the MJO related convection also better capture the SSW, but of course couldn't consider this particular event.

Thanks a lot for the insightful comment. We have taken a closer examination of the anomalies, using 4-day averages rather than weekly averages as mentioned in *Main topics point 2*. With the shorter window of time averaging, we do find anomalies, in particular the trough over the Northwestern Pacific, consistent with the impact associated with MJO phase 6/7 as suggested in Garfinkel et al. (2014); Liu et al. (2014). We have therefore included comments on the relevance to MJO at Lines 214 to 222.

line 187, 223/224: I don't see much evidence for a PNA pattern. Rather, there is a low in the far western Pacific and a ridge over Western North America. This is in-phase with wave-2, and hence constructively interferes with it. See e.g., Garfinkel et al 2010 (already cited) and Cohen and Jones 2011. Specifically for wave-2 the more relevant pattern is a low in Eastern Siberia and ridge over Alaska, and not a PNA per se.

Thanks, we agree that the PNA feature is not clearly shown in Figure 4. We have added a supplementary figure (Figure A2), which take a shorter time averaging of 4 days, compared to 7 days in Figure 4, in which the PNA pattern is better observed (Figure A2c). The mention of PNA pattern is now at 220 and we have also modified the text to state the relevance of a low in Eastern Siberia and ridge over Alaska in projecting wave-2 in Lines 207-208, 211-213.

line 192: "the North Pacific" -> "Alaska, but reduced SLP over Eastern Siberia,"

Thanks for the clarification and have modified, now at Line 211.

line 203-204 I found this sentence confusing. Please add some commas or rewrite

Thanks, have re-written and now at Line 229-230.

Rupp, P., Spaeth, J., Garny, H., and Birner, T.: Enhanced polar vortex predictability following sudden stratospheric warming events, Geophys. Res. Lett., 50, e2023GL104057, https://doi.org/10.1029/2023GL104057, 2023.

Spaeth, J., P. Rupp, H. Garny, T. Birner: Stratospheric impact on subseasonal forecast uncertainty in the Northern extratropics, Commun. Earth Environ., 5, 126, https://doi.org/10.1038/s43247-024-01292-z, 2024

Cho, H.-O., Kang, M.-J., Son, S.-W. (2023). The predictability of the 2021 SSW event controlled by the zonal-mean state in the upper troposphere and lower stratosphere. Journal of Geophysical Research: Atmospheres, 128, e2023JD039559. https://doi.org/10.1029/2023JD039559

Garfinkel, C. I., and C. Schwartz. "MJO-related tropical convection anomalies lead to more accurate stratospheric vortex variability in subseasonal forecast models." Geophysical research letters 44, no. 19 (2017): 10-054.

**Reviewer 3**

**Summary**

This analysis identified and examined the uncertainty in the stratospheric predictions tracking its origins and attempting to establish a link to the subsequent tropospheric uncertainties. The authors constructed two groups of ECMWF hindcasts with large and small spread in the zonal mean zonal wind at 60N 10hPa averaged over the whole length of the forecast. The results of the group comparison show that the main regions of the uncertainty source in the stratosphere are associated with the precursor regions in the troposphere: North Pacific and Northern Europe. This suggests that the tropospheric uncertainty is linked to the stratospheric one. Moreover, the authors studied the SSW that took place in February 2018 in more detail to identify the tropospheric responses and establish the two-way propagation of uncertainty. The show that the ensemble members that predicted the SSW correctly also better predicted its downward impacts. In my opinion this is a very compact, yet somewhat conclusive study that proves existing hypotheses and provides new insights into the stratospheric-tropospheric coupling. Therefore, I recommend this manuscript for publication, but I also suggest a few comments below.

**Major Comments**

- Averaging of the uncertainty over all 46-day integration period seems too broad and may filter out information. As the study focuses on the extended-range predictions it makes sense to average over week 3 and 4, for example. The authors may perform some sensitivity studies on that.

  Thanks for this comment. Along with the same concern from other reviewers, we have performed some sensitivity tests as mentioned in *Main topics point 1*. The 46-day averaging separates hindcasts into composites similar to using 10-46 or 23-46 day averages (i.e. week 3-4). The separation between the composites is largest at long lead times for all averaging periods. We therefore decided to keep the original 46-day definition and have added in Lines 91-93 a sentence stating that the results do not change significantly based on the change of averaging window.

- The section which addresses the downward coupling and study of the SSW event seems to be not very well connected and lacking evidence. For example, the regions of the upward wave propagation could be discussed in this case. Also, as the authors argue that the SSW cluster better predicted the downward impacts, it would be helpful to add some comparison to reanalysis to prove that point.

  Thank you for the comment. We have added supplementary figures (Figure A4 and A5) of eddy heat fluxes at 100hPa in the hindcast and reanalysis to highlight the regions of upward wave propagation. Figure A4 is mentioned at Line 212. We have also added a supplementary figure (Figure A6) illustrating the SSW downward impact in reanalysis. The hindcast successfully captures the downward impact of the SSW at long lead times but does differ from the reanalysis in the evolution of the anomalies associated with the downward impacts. We have extended the sentence commenting on the downward impact on Lines 225-227.

- The OLR analysis does not seem to explain the difference between the clusters in wave-2 activity. I suggest the authors add some more explanation of the link to the tropics.

  Thanks, we agree the OLR analysis is not sufficient to make the link. As mentioned in *Main topics point 2*, we have added supplementary figures in the manuscript (Figure A2 and A3) and provided more explanation on the link to the tropical convection on Lines 214-222.

**Minor Comments**

L34 I struggle to understand the part '..stationary waves, which interact with wave anomalies to determine the upward wave flux.' Please rewrite the sentence to make it clearer.

Thank you, we have modified the sentence, now on Lines 38-39.

L48 Please consider adding 'often resulting in better simulation of SSWs', as it is not always the case.

Thank you, we have modified the sentence, now on Line 53.

L52 Please consider adding (Statnaia and Karpechko, 2024) : a study investigating the overconfidence of the forecasts and its connection to the strength of the polar vortex.

Thank you, this is an interesting and relevant study and the reference is included, now at Lines 60 and 247.

L65 A new run for the 2018 SSW event is a very interesting addition to the study, but why you did not use the actual forecast issued on that date? Also, did you make only one run initialized on that date or several to choose from? It is very interesting that quite a few ensemble members predicted the event well in advance.

Thanks a lot for the comment, this is a very interesting point. We did several initializations to select the most fitting one. Specifically, we use a new run for the 2018 SSW event since the hindcast initialized on 2018-01-27 has larger ensemble spread and a larger number of ensemble members successfully predicting the event than the initializations available from the real-time forecast (A comparison is done in Figure R8). We have also added more explanations on the selection of this specific hindcast in Lines 74-76.

L102 Do you have the explanation for the significant $u_{1060}$ weakening after day 37 for the large $u_{1060}$ spread composite?

Thanks for the question and we suspect this is related to the wave activity that is still anomalously larger at long lead times compared to all hindcasts in Figure 1c. We have added some potential explanations at Line 114.

Figure 1 Only the ensemble mean for the 2018 SSW events seems to be plotted (purple dash line). Please correct the figure caption.

Thanks for the raising this point. The ensemble spread is plotted in Figure 1a. We have modified the figure caption to clarify.

References:

Statnaia, I. and Karpechko, A. Y. (2024) 'Factors influencing subseasonal predictability of northern Eurasian cold spells', Quarterly Journal of the Royal Meteorological Society, pp. 1–21. doi: 10.1002/qj.4744.

[Figure]

Figure R8: Ensemble plumes in zonal mean zonal wind at 10hPa, 60°N of ECMWF real-time forecasts initialized on (a) 2018-01-25 and (c) 2018-01-29, and of the re-run of the ECMWF hindcast initialized on (b) 2018-01-27. Lines in blue denote members that successfully predict the 2018 SSW event within 10 days following the SSW onset and in grey denote members tha did not predict the SSW. Numbers in the brackets at the legend indicate the number of ensemble members in each category. Ensemble spread in zonal mean zonal wind at 10hPa, 60°N averaged over the entire hindcast/ forecast period is denoted on the bottom right hand corner at each panel. Lag 0 denotes the onset of the 2018 SSW.

**References**

Garfinkel, C. I., Benedict, J. J. and Maloney, E. D. (2014), 'Impact of the MJO on the boreal winter extratropical circulation', *Geophys. Res. Lett.* **41**(16), 6055–6062.

Garfinkel, C. I., Butler, A. H., Waugh, D. W., Hurwitz, M. M. and Polvani, L. M. (2012), 'Why might stratospheric sudden warmings occur with similar frequency in El Niño and La Niña winters?', *J. Geophys. Res. Atmos.* **117**(D19).

Kent, C., Scaife, A. A., Seviour, W. J. M., Dunstone, N., Smith, D. and Smout-Day, K. (2023), 'Identifying Perturbations That Tipped the Stratosphere Into a Sudden Warming During January 2013', *Geophys. Res. Lett.* **50**(24), e2023GL106288.

Liu, C., Tian, B., Li, K.-F., Manney, G. L., Livesey, N. J., Yung, Y. L. and Waliser, D. E. (2014), 'Northern Hemisphere mid-winter vortex-displacement and vortex-split stratospheric sudden warmings: Influence of the Madden-Julian Oscillation and Quasi-Biennial Oscillation', *J. Geophys. Res. Atmos.* **119**(22), 12,599–12,620.

Schwartz, C., Garfinkel, C. I., Yadav, P., Chen, W. and Domeisen, D. I. V. (2022), 'Stationary wave biases and their effect on upward troposphere– stratosphere coupling in sub-seasonal prediction models', *Weather Clim. Dyn.* **3**(2), 679–692.

---

## Author Response (AR2)

**Response to the reviewers of egusphere-2024-1652: Tropospheric Links to Uncertainty in Stratospheric Subseasonal Predictions**

Dear Editor Petr Šácha,

On behalf of all authors I would like to submit the revised version of the original article "Tropospheric Links to Uncertainty in Stratospheric Subseasonal Predictions" including an annotated version of the modified manuscript.

Best regards,
Rachel Wu
* * *
Dear authors,

thank you very much for carefully considering all of the referee comments, which deserve credit for improving the manuscript considerably. I am happy to write you that I reached the editorial decision of "Publish subject to minor revisions" . Also, I confirm the intention to nominate your article for a highlight paper. The minor revisions that I recommend below should help you to reach this goal and are mainly technical in nature. Definitely, they do not prevent publication of your manuscript.

Thank you for carefully reviewing our revision of the manuscript and the intention to nominate our article for a highlight paper. We have addressed the revisions as recommended. Please find below the detailed responses (in blue) to the reviewers' comments and suggestions. All line indications refer to the new (annotated) version of the manuscript.

Editorial comments (numbering follows the tracked changes version):

1) At numerous places of the manuscript (L94, L108, L171, L252), the not shown statement is invoked. I feel that particularly in the conclusions this is not appropriate, as here the most important findings supported by the results presented should be discussed. The manuscript is quite short right now and very readable, which I like very much. On the other side, I feel that there may be some space for adding a figure or two (from the Appendix?) to the main text and/or for expanding the Appendix (e.g. by the figures from the review process) to decrease the number of not shown statements. I would like to ask you to consider this comment carefully, seeking balance between readability on the one side and self-consistency on the other.

Thank you for the suggestion. We have now added Figure A1 and A2 to reduce the number of 'not shown' statements. Figure A1 is referred to on Line 75. Figure A2 is referred to on Lines 107, 171, 176 and 178.

2) In Fig.3, Fig. 5 (and in Figs. Ax) you cut the plots at around 60°S. While I understand your motivation, this results only in small space savings. Moreover, it seems that the significance regions would often extend to the SH polar region, if shown, challenging either your methodology or our current understanding. If you decide to keep the figures as they are, please include a statement justifying this cut-off.

Thank you for the suggestion. We have extended our plots to include the Southern Hemisphere (Figure 3, 5 and A4). Regions of significance are found over the Southern Hemisphere and are potentially related to tropical precursors. We have added some description on Lines 154-159 and 215-217.

Technical:

Chronological ordering (numbering follows the tracked changes version):

L26 - depositing wave momentum -> depositing momentum..this is more precise, because strictly speaking there is nothing like wave momentum, only wave momentum flux..

*Thank you for the comment. The phrase is corrected on Line 26.*

L60 - the phrase in particular appears twice in a short sequence, consider rephrasing

*Thank you for the comment. 'In particular' is replaced by 'especially' on Line 59.*

L93 - a typo - the results are do not change...

*Thank you for spotting the typo. The phrase is now corrected on Line 91.*

L166 - (Figure 2f) (Schwartz et al., 2022)...consider rephrasing. It is not clear, whether you point the reader to a possible Fig. 2f in Schwartz et al.

*Thank you for the comment. The phrase is modified on Line 169.*

L170-L172 - Please clearly state again whether this can be related to the positive impact of early SSW occurrence on the subsequent spread (downward impacts from the stratosphere).

*Thank you for the comment. The results and sensitivity tests show that it is not possible to clearly separate tropospheric precursors and responses in uncertainty when considering all hindcasts together, as noted in the Conclusions (Lines 259-261). We have therefore added sentences elaborating on the timing of SSW occurrences in the hindcast composites and clarified that the overlap in timing makes it difficult to separate the direction of coupling. These clarifications are found at Lines 170-171 and 177-181.*

L184-L187 I cannot see the extreme spread for the purple dashed line in Fig. 1a, nor the strong vortex state in Fig. 1b. Can you possibly clarify or guide the reader's eye better?

*Thank you for spotting this. We realised we have accidentally plotted the wrong purple lines while updating the figure during the last revision. We have restored the figure to the version at the initial submission. The purple dashed line in Fig. 1a now shows extreme spread, and the purple dashed line in Fig. 1b shows the strong vortex state at initialization.*

Starting at page 13: Please double check the formatting of the Appendix. It seems to be scattered between the Data availability and Acknowledgements statements.

*Thank you for the comment. The data availability and acknowledgement statements are now all after the Appendix section.*

Thank you very much again for publishing with ACP and wishing you all the best for the next steps.

Best regards,

Petr Šácha.